# IF-VIDCAP: CAN VIDEO CAPTION MODELS FOLLOW INSTRUCTIONS?

**Shihao Li[1*], Yuanxing Zhang[2*], Jiangtao Wu[1*], Zhide Lei[1*], Yiwen He[1], Runzhe Wen[1], Chenxi Liao[1], Chengkang Jiang[1], An Ping[1], Shuo Gao[1], Suhan Wang[1], Zhaozhou Bian[1], Zijun Zhou[4], Jingyi Xie[4], Jiayi Zhou[1], Jing Wang[1], Yifan Yao[1], Weihao Xie[5], Yingshui Tan[5], Yanghai Wang[1], Qianqian Xie[1], Zhaoxiang Zhang[3], Jiaheng Liu[1,†]**

[1]Nanjing University, [2] Kuaishou Technology, [3]CASIA, [4]Shanghai University, [5]M-A-P
lishihao@smail.nju.edn.cn    liujiaheng@nju.edn.cn

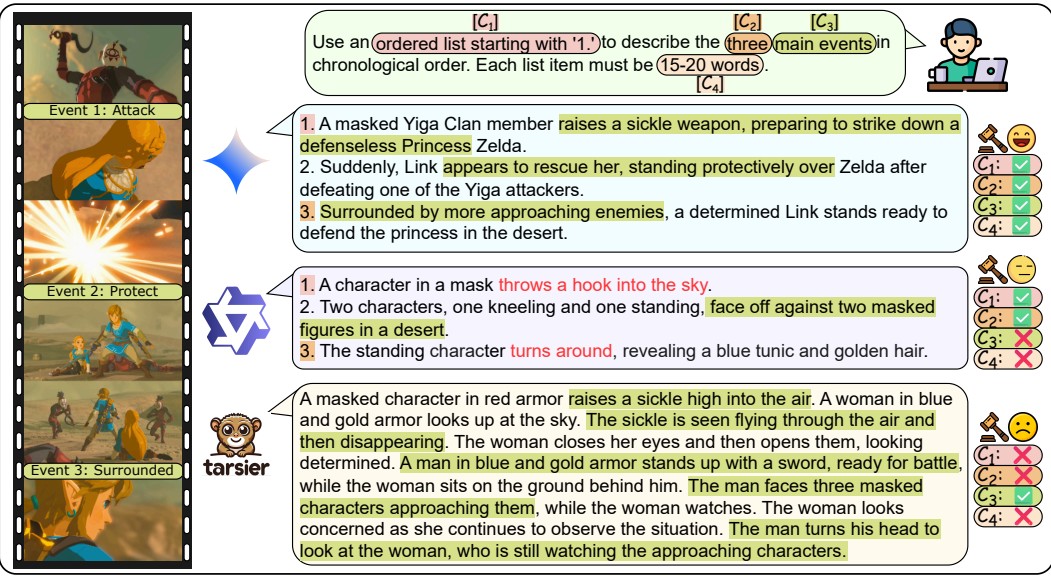

Figure 1: Differences in Controlled Video Captioning Capabilities among MLLMs. In our case, constraint types are color-coded, with each color corresponding to a category in Figure 3d, and sentences highlighted in red indicate incorrect event captions.

## ABSTRACT

Although Multimodal Large Language Models (MLLMs) have demonstrated proficiency in video captioning, practical applications require captions that follow specific user instructions rather than generating exhaustive, unconstrained descriptions. Current benchmarks, however, primarily assess descriptive comprehensiveness while largely overlook instruction-following capabilities. To address this gap, we introduce IF-VidCap, a new benchmark for evaluating controllable video captioning, which contains 1,400 high-quality samples. Distinct from existing video captioning or general instruction-following benchmarks, IF-VidCap incorporates a systematic framework that assesses captions on two dimensions: format correctness and content correctness. Our comprehensive evaluation of 26 prominent models reveals a nuanced landscape: despite the continued dominance of proprietary models, the performance gap is closing, with top-tier open-source solutions now achieving near-parity. Furthermore, we find that models specialized for dense captioning underperform general-purpose MLLMs on complex instructions, indicating that future work should simultaneously advance both descriptive richness and instruction-following fidelity. [1]

---

* Equal contribution.  [†] Corresponding Author.
[1]https://if-vidcap.github.io/

# 1 INTRODUCTION

High-quality, controllable video captions are crucial for a range of downstream tasks, including structured captions for video generation (Ju et al., 2024), targeted descriptions for video editing (Jiang et al., 2025), and stylistically appropriate copy for content creation (Ye et al., 2024). While recent Multimodal Large Language Models (MLLMs) excel at general video description (Yuan et al., 2025; Wang et al., 2025a), they often struggle to adhere to fine-grained instructions (Lian et al., 2025) that impose specific stylistic or formatting constraints. As illustrated in Figure 1, this limitation highlights a critical gap between their strong perceptual abilities and their weaker fidelity to complex user directives. We argue this gap stems from the fact that such tasks require not just perception, but a sophisticated synthesis of reasoning and constrained generation.

Existing evaluation for multimodal understanding, which primarily consists of question-answering (Fu et al., 2025; Zhou et al., 2025; Li et al., 2024) and captioning tasks (Chai et al., 2025; Wang et al., 2024; Chen et al., 2025), lacks the sophisticated instruction-following assessments common in language-only benchmarks (Zhou et al., 2023; Zhang et al., 2025c). Specifically, benchmarks on video captioning traditionally prioritize descriptive accuracy and comprehensiveness, often neglecting practical constraints such as output format, length, and adherence to specific content requirements or prohibitions. As controllable video captions for diverse downstream applications become more prevalent, **assessing a MLLM's capacity to fulfil varied structural and semantic instructions within a single generation has emerged as a critical challenge**.

To evaluate fine-grained instruction following in video captioning, we introduce IF-VidCap, a new benchmark comprising 1,400 instructions across over 13 video categories. These instructions incorporate 27 distinct constraint types—such as output formatting, aspect restrictions, and element focus—and feature high compositional complexity, with an average of six constraints each. We implement the benchmark with an automated evaluation protocol that leverages high-quality annotations to assess both instruction fidelity and semantic quality, as illustrated by Figure 2. Experiments on 26 state-of-the-art models demonstrate the benchmark's difficulty and discriminative power, revealing a significant performance gap in which closed-source models generally outperform their open-source counterparts. The main contributions are summarized as follows:

- **The first instruction-following video captioning benchmark**. We propose a new, high-quality benchmark, IF-VidCap, featuring 1,400 complex, compositional instructions aligned with realworld downstream applications. IF-VidCap could push the boundaries of instruction-following evaluation for video captioning.

- **A robust evaluation protocol to evaluate instruction fidelity and semantic quality**. IF-VidCap is attached with a robust, multi-dimensional evaluation protocol that combines both rule-based and LLM-based checks. The annotations on all samples within the evaluation have been carefully checked for coverage, correctness, and stability.

- **Discrimination of mainstream MLLMs and insights on the instruction-following video captioning**. We observe significant performance disparities among current MLLMs, particularly highlighting the poor performance of specialized captioning models when faced with instructional constraints. These findings suggest that the integration of captioning and instruction-following abilities could be a critical direction for future research.

- **A Training Dataset for Instruction-Following**. To enable fine-grained, instruction-based control, we introduce a new training dataset that we have specifically curated. We use this data to fine-tune the Qwen2.5-VL-7B-Instruct model. The resulting model, IF-Captioner-Qwen, achieves improved instruction-following performance over the base model.

# 2 RELATED WORKS

**Instruction-Following.** The evaluation of instruction-following for Large Language Models (LLMs) has matured significantly, evolving alongside the models' capabilities (Ouyang et al., 2022). Early benchmarks focused on general task performance (Jiang et al., 2024; Qin et al., 2024b) or programmatic verifiability (Zhou et al., 2023). More recently, the focus has shifted towards assessing adherence to complex, compositional instructions (Zhang et al., 2025c; Wen et al., 2024) and specialized domain constraints (He et al., 2024; Qin et al., 2024a). This emphasis is critical, as training

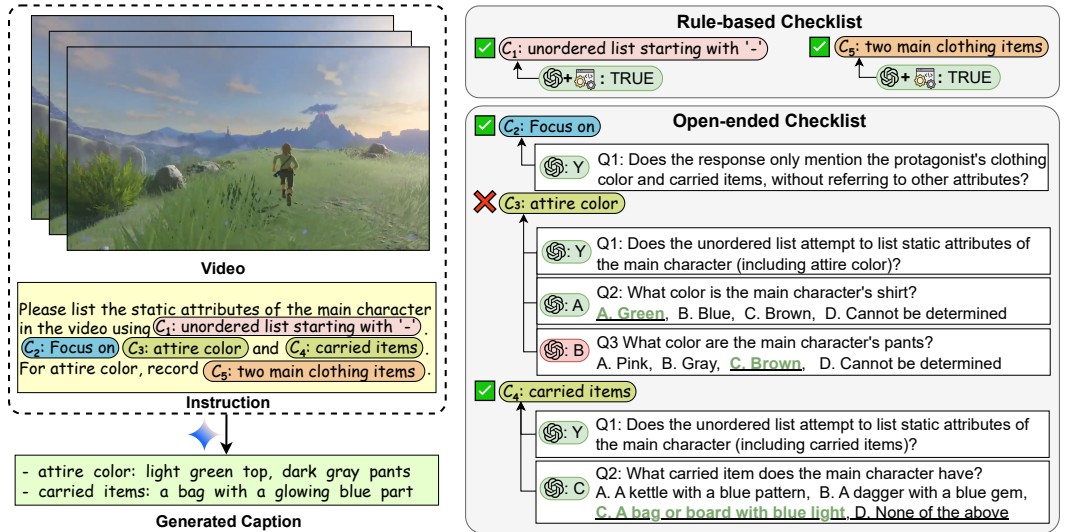

Figure 2: Sample data in IF-VidCap. The constraint types are color-coded, with colors corresponding to the categories in Figure 3d. Our checklist is divided into two types based on the checking method: rule-based items checked by LLM with rule scripts and open-ended items checked by LLM. The rule-based items cover format correctness, while the open-ended items cover semantic and content correctness.

on such intricate instructions demonstrably improves model fidelity (Sun et al., 2024; Mukherjee et al., 2023). However, this rigorous evaluation paradigm has remained largely confined to text-only tasks. Existing multimodal benchmarks, by contrast, tend to prioritize broad cross-task generalization (Bitton et al., 2023) over fine-grained instructional diversity within a single domain. IF-VidCap directly addresses this gap by introducing a systematic, instruction-centric evaluation framework for the foundational task of video captioning.

**Video Captioning Benchmarks.** Traditional video captioning benchmarks are designed to evaluate the semantic quality of generated text, not a model's adherence to arbitrary instructions. They typically fall into several categories: those targeting fine-grained descriptions (Chen et al., 2025; Wang et al., 2024; Zhang et al., 2025b), those assessing structured or scripted outputs (Ju et al., 2024; Yang et al., 2024), and those focused on specific applications like content retrieval (Xu et al., 2025) or long-form video understanding (Ren et al., 2024). While crucial for advancing descriptive capabilities, these benchmarks share a common limitation: they evaluate against a fixed set of quality criteria, such as accuracy or detail, rather than dynamic user-defined constraints. In contrast, IF-VidCap is the first benchmark in video captioning designed to systematically measure a model's ability to comprehend and execute diverse and compositional instructions.

## 3 IF-VIDCAP

To systematically address the challenge of controllable video captioning, our first step was to analyze the instructional requirements of diverse downstream applications like video editing and content creation. From this analysis, we distilled a comprehensive constraint framework encompassing 27 distinct types (Figure 3d), which serves as the blueprint for our benchmark and ensures its relevance to practical needs.

Based on this framework, IF-VidCap is built on a Video-Instruction-Checklist triplet. After curating high-quality videos, we employed a hybrid generation-and-refinement pipeline to create the dataset. Each checklist features two components for evaluation: rule-based items to assess hard constraints like format and structure, and open-ended QA to probe semantic fidelity. This paradigm provides a comprehensive assessment of a model's controllability.

## 3.1 DATA COLLECTION

### 3.1.1 VIDEO COLLECTION

To construct a high-quality evaluation benchmark, we selected 350 videos to build the test set, began by assembling a large, copyright-free video pool from academic datasets and public platforms. Each video was then subjected to a filtering process, disqualifying content based on technical deficiencies (e.g., resolution below 480p, duration outside the 2–60s range), poor visual composition (e.g., excessive clutter), or data integrity issues (e.g., duplicates). The resulting collection offers extensive variety—from animation to natural landscapes—ensuring the benchmark is reliable, and free from common data quality pitfalls.

### 3.1.2 ANNOTATION PIPELINE

Our annotation pipeline employs a two-stage workflow that combines automated generation with expert validation to achieve both scale and quality.[2]

**Stage 1:** Automated Draft Construction. An Instruction Generator produces instruction–checklist pairs for each video. Subsequently, a suite of Response Generators creates multiple candidate captions, which are assessed by an Automated Evaluator to provide preliminary judgments.

**Stage 2:** Human Refinement and Validation. A team of professionally trained annotators meticulously refines the automatically generated draft, resulting in an 83.6% modification rate. Consensus among all three annotators was necessary for a sample's acceptance; any conflicts in opinions were resolved by a senior supervisor. This process yielded the final 1,400 high-quality samples.

## 3.2 DATASET STATISTICS

### 3.2.1 OVERALL STATISTICS

Statistical analysis of the IF-VidCap dataset confirms its suitability as a comprehensive benchmark, distinguished by its diverse duration, content coverage, and instructional complexity (Figure 3d). The dataset features a uniform distribution of video durations, which are, on average, longer than those in existing short-video benchmarks (Figure 3a). Its broad content diversity, spanning numerous categories (Figure 3b), facilitates robust testing of cross-domain generalization. Furthermore, instruction complexity ranges from standard prompts (3–8 constraints) to highly challenging cases (over 10 constraints), specifically designed to probe compositional reasoning (Figure 3c). Together, these characteristics establish IF-VidCap as a next-gen testbed for evaluating MLLMs.

### 3.2.2 COMPARISON WITH OTHER BENCHMARKS

We adopt IFEval (Zhou et al., 2023), CELLO (He et al., 2024), InfoBench (Qin et al., 2024b), FollowBench (Jiang et al., 2024), SysBench (Qin et al., 2024a) , CFBench (Zhang et al., 2025c) and ComplexBench (Wen et al., 2024) as the instruction-following baselines.For captioning, we use CapsBench (Liu et al., 2024), Dream-1k (Wang et al., 2024) and CaReBench (Xu et al., 2025).As shown in Table 1, IF-VidCap advances both Instruction Following and Captioning benchmarks. Unlike prior text-only IF datasets, it introduces the video modality while offering greater scale (1,400 samples), complexity (6 avg. constraints), and broader content diversity. For video captioning, it pioneers a shift from dense description to fine-grained instruction following, featuring more challenging longer content with longer videos (20.5s). By bridging these domains, IF-VidCap provides a more rigorous benchmark for evaluating controllable generation in multimodal models.

## 3.3 EVALUATION PROTOCOL

### 3.3.1 EVALUATION CRITERIA

IF-VidCap evaluates two core dimensions—instruction following and video description quality—employing LLM-driven methods to ensure both flexibility and scalability.

---

[2]The detailed process for constructing the test set is provided in Appendix D.

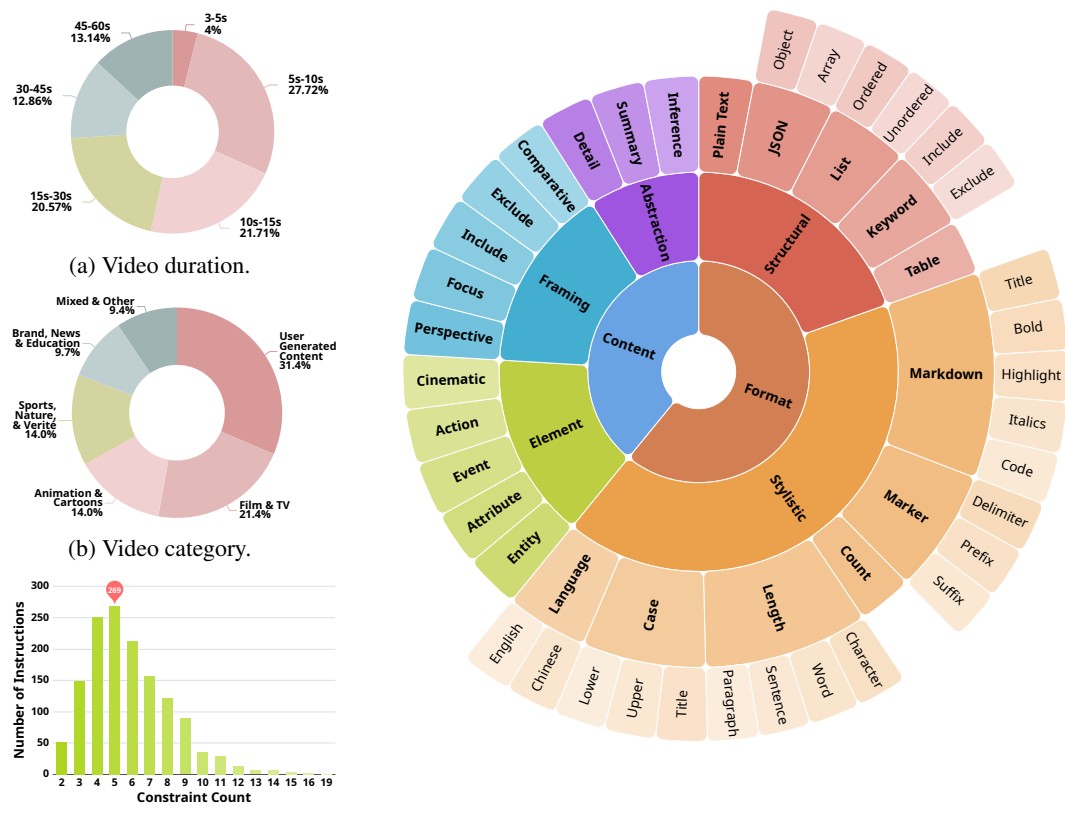

(a) Video duration.

(b) Video category.

(c) Constraint count.

(d) Overview of IF-VidCap constraint categories.

Figure 3: Dataset statistics for IF-VidCap. (a-c) show distributions for video duration, category, and constraint count, respectively. (d) provides an overview of the constraint categories.

Table 1: Comparison of Instruction Following and Video Captioning Benchmarks. "#Size", "#Types", "#Const.", "Vid. Len.", and "#QAs" denote the number of samples, constraint types, average constraints per instruction, average video length, and average QA pairs per caption, respectively. "Mod." indicates the input modality (text/image/video), while "Evaluation" specifies the evaluation method (Rule-based, LLM-as-Judge, or a combination).

| Benchmark | #Size | #Types | #Const. | Vid. Len. | #QAs | Mod. | Evaluation |
|---|---|---|---|---|---|---|---|
| *Instruction Following Benchmarks* | | | | | | | |
| IFEval | 541 | 25 | 1.54 | – | – | Text | Rule |
| CELLO | 523 | 4 | 2.18 | – | – | Text | Rule |
| InfoBench | 500 | 5 | 5.93 | – | – | Text | LLM |
| FollowBench | 944 | 5 | 3.0 | – | – | Text | LLM / Rule |
| SysBench | 500 | 6 | 2.38 | – | – | Text | LLM |
| CFBench | 1,000 | 10-25 | 4.24 | – | – | Text | LLM |
| ComplexBench | 1,150 | 4-19 | 4.61 | – | – | Text | LLM+Rule |
| *Captioning Benchmarks* | | | | | | | |
| CapsBench | 200 | – | – | – | 12.36 | Image | LLM |
| VidCapBench | 643 | – | – | 10.25s | 16.55 | Video | LLM |
| Dream-1K | 1,000 | – | – | 8.9s | – | Video | LLM |
| CaReBench | 1,000 | – | – | 14.35s | – | Video | LLM |
| **IF-VidCap (Ours)** | 1,400 | 27 | 6.00 | 20.5s | 16.95 | Video | LLM+Rule |

**Instruction Following Evaluation**: The evaluation paradigm has shifted toward using LLMs as automated judges. Common strategies include pairwise comparison for estimating win rates, direct assessment for computing pass rates, and rule-augmented systems(Wen et al., 2024). The rule-augmented approach, which combines LLM-based semantic extraction with deterministic checks, is

particularly effective for complex instructions, leveraging both the nuanced generalization of LLMs and the reliability of rule-based verification.

**Video Captioning Evaluation**: Evaluation criteria have evolved toward semantic fidelity. While structured matching (Chai et al., 2025; Wang et al., 2024) and rubric scoring (Ren et al., 2025) outperform traditional metrics, they often fall short in capturing fine-grained details. In contrast, Question-Answering (QA) provides a more rigorous framework to validate such nuances.

Building on the above insights, IF-VidCap adopts a composite mechanism for constraint satisfaction evaluation(Figure 2). For rule-checkable constraints, we combine an LLM with rule scripts: the LLM serves as a content extractor while the rule scripts act as the verification executor. This integrates the LLM's adaptability to complex text processing with the determinism of rule execution. For open-ended constraint checking, we design retrieval-based QA pairs, enabling the LLM to answer using the video caption as context. Specifically, the checklist uses true/false questions to let the LLM directly judge the semantic correctness of the description, and multiple-choice questions to have the LLM select facts inferable from the video description. All answers provided by the LLM are compared against the ground truth, and statistics are aggregated at the constraint level (a single constraint may include multiple QA pairs to enable atomic checking and control over granularity).

### 3.3.2 Evaluation Metrics

Our evaluation metrics are based on the following two main instruction-following evaluation metrics:Constraint Satisfaction Rate (CSR) and Instruction Satisfaction Rate (ISR).

$$\text{CSR} = \frac{1}{m} \sum_{i=1}^{m} \frac{1}{n_i} \sum_{j=1}^{n_i} s_i^j, \qquad \text{ISR} = \frac{1}{m} \sum_{i=1}^{m} s_i \qquad (1)$$

where $s_i^j = 1$ if the $j$-th constraint of the $i$-th instruction is satisfied, otherwise 0; $s_i = 1$ if all constraints of the $i$-th instruction are satisfied, otherwise 0; $m$ denotes the total number of instructions, and $n_i$ represents the total number of constraints for the $i$-th instruction.

Additionally, based on the evaluation system of IF-VidCap, we propose more fine-grained metrics:

- **Rule-Based CSR/ISR**: Only considers format-related constraints, which reflects the model's ability to control output format.

- **Open-Ended CSR/ISR**: Only considers content-related constraints, which reflects the model's ability to understand and describe multimodal content.

### 3.4 Model Training

Furthermore, we introduce a fine-tuning dataset designed to teach models generalizable instruction-following skills. Critically, its generation process is intentionally distinct from our test set's. We began by sourcing 11K high-quality video-caption pairs from datasets including Vript (Yang et al., 2024), ShareGPT4Video (Chen et al., 2024), VideoUFO-Gemini (Wang & Yang, 2025), and ShareGPT-4o (Cui et al., 2024). For each pair, we adopted a "response-to-instruction" approach: treating the existing caption as a textual proxy for the video's content, we prompted DeepSeek-V3.1 to synthesize diverse instructions based on our constraint framework[3]. This leverages the LLM's pre-trained knowledge to mirror varied, real-world task preferences. To prevent stylistic overfitting, the prompts used in this process are entirely different from those for our test set.

Ultimately, this process yields 11K curated video-caption pairs, from which we construct a training set of 46K video-instruction-response triplets to fine-tune the Qwen-2.5-VL-7B-Instruct model.

---

[3]The complete details of the constraint framework are available in Appendix E

## 4 EXPERIMENTS

### 4.1 MAIN RESULTS

We evaluate 26 popular models including Gemini-2.5-Pro/Flash (Team, 2025), GPT-4o (OpenAI, 2024), InternVL-3.5 (Wang et al., 2025a), Qwen2.5-VL-Instruct (Bai et al., 2025), VideoL-LaMA3 (Zhang et al., 2025a), MiniCPM-V-4.5 (Yao et al., 2024), Llama-3.2-Vision-Instruct (Lee et al., 2025), LLaVA-V1.6-Vicuna (Liu et al., 2023), Video-LLaVA (Lin et al., 2024), ARC-Hunyuan-Video (Kong et al., 2025), MiMo-VL (Team et al., 2025a), Kimi-VL (Team et al., 2025b), LLaVA-Video (Zhang et al., 2025d), VideoChat2 (Li et al., 2024),InternVideo2.5 (Wang et al., 2025b), GLM-4.1V (Team et al., 2025c) and Tarsier (Yuan et al., 2025).

The main results in Table 2 lead to the following key observations: (1) Performance scales with model size within the same family. (2) Top open-source models now rival closed-source counterparts. (3) The "Thinking" mode's superior performance underscores the necessity of reasoning for complex tasks. (4) Instruction Satisfaction Rate (ISR) is inherently lower than Constraint Satisfaction Rate (CSR), as it requires concurrently meeting all constraints. (5) Models exhibit stronger control over format (rule-based checks) than content (open-ended checks), likely because content requires complex multi-modal reasoning, whereas format is often text-based. (6) The human baseline exhibits distinct behavioral differences compared to advanced models. Since the human reference relies on 10 untrained undergraduates (whose independent results were merged), their performance in open-ended content description slightly trails behind top-tier models. Conversely, driven by human mechanisms of careful checking and self-reflection, they demonstrate superior performance in rule-based format control, substantially surpassing all models. Interestingly, the positive effect of such reflection on enforcing format constraints is mirrored in "Thinking" models, as their Chain-of-Thought (CoT) often incorporates an explicit final check and correction of the format.

Additionally, we trained an IF-Captioner-Qwen model, which was created by fine-tuning Qwen2.5-VL-7B-Instruct on our self-constructed dataset[4]. We observe that IF-Captioner-Qwen outperforms Qwen2.5-VL-7B-Instruct by a large margin on both ISR and CSR metrics.

### 4.2 FURTHER ANALYSIS

**Effect of Different Instruction Metrics.** We analyze the evaluation results of four representative models to investigate how Constraint Satisfaction Rate (CSR) and Instruction Satisfaction Rate (ISR) vary with instruction complexity, which we define jointly by constraint count and prompt length. Experiments are conducted on a curated test set of 1021 high-quality instances, expert-selected from the full benchmark. In these instances, constraints exhibit dependencies (e.g., chaining, nesting, choice) rather than being mutually independent. This selection ensures that constraint count serves as a reliable proxy for overall complexity. The results (Figure 4a and 4b) clearly demonstrate that a model's ability to satisfy constraints and adhere to instructions degrades as instruction complexity increases, confirming that models' instruction-following capabilities are significantly challenged by more intricate and difficult commands.

**Effect of Different Video Parameters.** We analyze the sensitivity of the Qwen2.5-VL-7B-Instruct model to different input video parameters. Fixing the resolution at 224×224 and varying frame counts (8, 16, 32, 64, 128), ISR and CSR generally increased with more frames, peaking at 64 before dropping at 128—likely due to the model's limited capacity for very long sequences (Figure 4c). With a constant 32-frame sequence, increasing resolution from 168×168 to 784×784 consistently improved both metrics (Figure 4d). Overall, the model benefits from richer temporal context and higher spatial fidelity, though excessively long sequences can harm performance.

**Agreement Evaluation.** To assess the alignment between our evaluation method and human judgment, we use the professional annotations obtained in Section 3.1.2 as the human reference. Across the entire test set, we measured the agreement between automated evaluations from three different assessor models and the human evaluations. The assessor models include GPT-5-mini (OpenAI, 2025), DeepSeek-V3.1-NoThink (DeepSeek-AI et al., 2025) and Qwen3-32B (Yang et al., 2025). In Table 3, the overall agreement of the automated evaluation is strong—particularly for the advanced

---

[4]The detailed training process and hyperparameter settings are provided in Appendix H

Table 2: Results of different models. 💡 indicates that models support the "thinking" mode, and the scores split by "/" denote the performance of thinking and non-thinking modes, respectively.

| Models | Params | Overall | | Rule-based | | Open-ended | |
|---|---|---|---|---|---|---|---|
| | | ISR | CSR | ISR | CSR | ISR | CSR |
| Human | ◑ | 31.89 | 75.57 | 78.25 | 93.64 | 33.93 | 55.82 |
| *Closed-Source Large Multimodal Models* | | | | | | | |
| Gemini-2.5-Pro | 🔒 | 27.83 | 74.53 | 74.35 | 87.81 | 35.22 | 59.00 |
| Gemini-2.5-Flash | 🔒 | 25.50 | 72.63 | 67.80 | 84.51 | 35.45 | 58.71 |
| GPT-4o | 🔒 | 22.90 | 70.74 | 69.20 | 85.12 | 30.94 | 53.91 |
| Gemini-2.0-Flash | 🔒 | 18.19 | 67.45 | 63.04 | 82.06 | 26.86 | 50.39 |
| *Open-Source Large Multimodal Models* | | | | | | | |
| Qwen3-VL-Instruct | 235B | 26.41 | 71.65 | 67.16 | 84.14 | 36.39 | 57.12 |
| InternVL-3.5💡 | 241B | 24.20 | 71.17 | 65.58 | 83.21 | 34.64 | 57.13 |
| InternVL-3.5💡 | 38B | 20.71 / 15.43 | 68.30 / 64.76 | 59.43 / 57.79 | 80.17 / 78.92 | 31.79 / 24.93 | 54.42 / 48.20 |
| Mimo-VL-SFT | 7B | 17.72 | 66.33 | 56.2 | 78.14 | 28.96 | 52.54 |
| Qwen2.5-VL-Instruct | 72B | 17.50 | 67.28 | 64.29 | 83.22 | 25.71 | 48.65 |
| InternVL-3.5💡 | 8B | 17.33 / 9.96 | 65.90 / 56.45 | 60.32 / 48.14 | 79.95 / 71.68 | 26.84 / 16.98 | 49.50 / 38.65 |
| Qwen2.5-VL-Instruct | 32B | 15.16 | 64.04 | 53.66 | 76.95 | 26.72 | 48.94 |
| VideoLLaMA3💡 | 7B | 12.21 / 10.63 | 57.38 / 57.17 | 48.64 / 47.34 | 71.69 / 71.21 | 19.93 / 18.46 | 40.65 / 40.75 |
| MiniCPM-V-4.5💡 | 8B | 11.75 / 8.57 | 61.67 / 59.23 | 58.09 / 56.07 | 79.35 / 77.62 | 18.05 / 14.64 | 40.97 / 37.73 |
| GLM-4.1V | 9B | 11.46 | 57.88 | 47.64 | 70.48 | 20.06 | 43.14 |
| Qwen2.5-VL-Instruct | 7B | 10.92 | 58.12 | 52.51 | 73.81 | 18.75 | 39.65 |
| Kimi-VL-Instruct | 16B | 9.29 | 53.94 | 40.21 | 63.73 | 19.86 | 42.5 |
| LLaVA-Video-Qwen2 | 7B | 8.93 | 53.43 | 41.86 | 65.59 | 17.64 | 39.22 |
| VideoChat2-HD-stage4-Mistral | 7B | 8.82 | 50.67 | 52.06 | 68.51 | 13.82 | 27.22 |
| Internvideo2.5 | 7B | 7.42 | 51.54 | 41.43 | 65.57 | 13.54 | 35.16 |
| Qwen2.5-VL-Instruct | 3B | 6.54 | 51.74 | 43.46 | 66.50 | 13.15 | 34.47 |
| Llama-3.2-Vision-Instruct | 90B | 5.80 | 45.18 | 36.03 | 59.56 | 11.03 | 28.36 |
| Llama-3.2-Vision-Instruct | 11B | 4.00 | 39.87 | 31.29 | 53.24 | 7.71 | 24.25 |
| LLaVA-V1.6-Vicuna | 7B | 3.54 | 43.92 | 35.84 | 60.09 | 7.30 | 25.02 |
| Video-LLaVA | 7B | 3.13 | 38.74 | 26.53 | 51.27 | 7.73 | 24.05 |
| ARC-Hunyuan-Video | 7B | 2.32 | 27.78 | 12.23 | 31.41 | 9.11 | 23.54 |
| Tarsier2 | 7B | 1.40 | 26.05 | 9.30 | 27.75 | 9.91 | 24.04 |
| **IF-Captioner-Qwen (Ours)** | 7B | 14.63 | 62.82 | 59.13 | 79.03 | 21.27 | 43.99 |

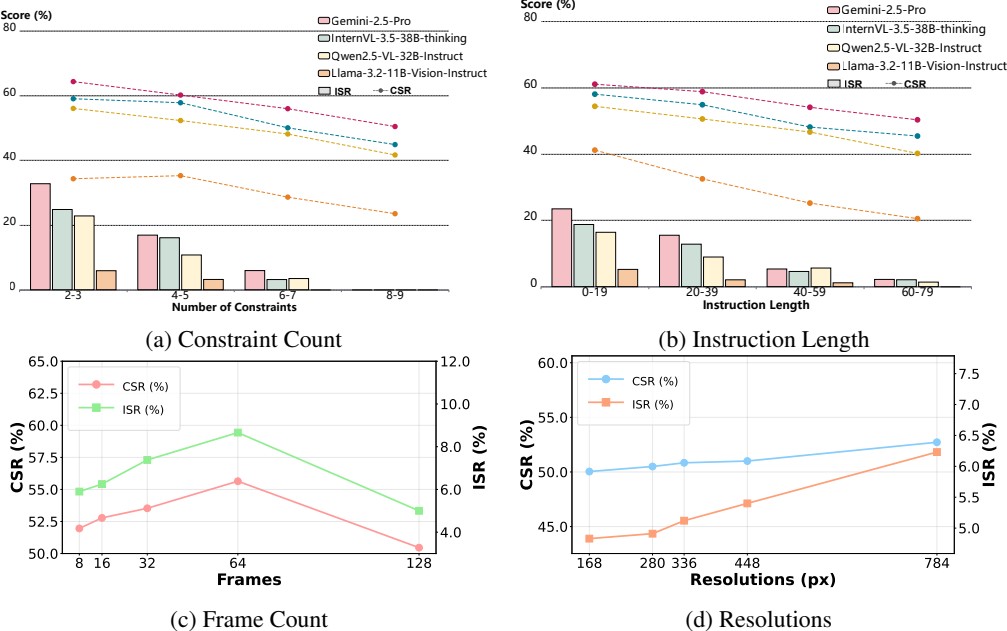

(a) Constraint Count

(b) Instruction Length

(c) Frame Count

(d) Resolutions

Figure 4: The Impact of Constraint Count, Instruction Length, Frame Count, and Resolutions on Instruction-Following Capability (CSR and ISR).

proprietary evaluator (GPT-5-mini), which reaches 96.33%—thereby validating the reliability of our evaluation pipeline. We also provide a scored-based consistency analysis fo CSR/ISR in Table 4, both open and closed-source models demonstrate a high degree of consistency. For open-source

models, the agreement is slightly weaker yet still considerable, indicating that the evaluation method is broadly applicable across different types of models.

Table 3: Agreement between automated evaluation and human evaluation across different models.

| Model | Overall Agreement | Rule-based. | Open-ended. |
|---|---|---|---|
| GPT-5-mini | 96.33 | 96.90 | 96.08 |
| Qwen3-235B-A22B-Instruct | 94.78 | 95.35 | 94.24 |
| DeepSeek-V3.1-NoThink | 92.18 | 93.55 | 91.58 |
| Qwen3-32B | 92.03 | 92.10 | 92.00 |

Table 4: Agreement between automated evaluation and human evaluation across different models with ISR/CSR

| Model | Overall | | Rule-based | | Open-ended | |
|---|---|---|---|---|---|---|
| | ISR | CSR | ISR | CSR | ISR | CSR |
| GPT-5-mini | 23.86 | 69.45 | 63.57 | 82.92 | 33.79 | 55.92 |
| Qwen3-235B-A22B-Instruct | 22.53 | 69.36 | 59.39 | 81.25 | 36.35 | 58.02 |
| DeepSeek-V3.1-NoThink | 22.50 | 68.95 | 56.21 | 78.04 | 35.36 | 57.28 |
| Qwen3-32B | 17.25 | 65.10 | 54.27 | 77.68 | 28.01 | 50.02 |
| Human | 23.89 | 69.57 | 63.00 | 82.64 | 33.93 | 55.82 |

**Constraint Type Analysis.** Our statistical analysis of the Constraint Satisfaction Rate (CSR) for a set of representative models across the entire benchmark (see Figures 5) reveals that current multimodal models exhibit distinct preferences when adhering to different types of constraints.

Overall, general-purpose models excel at format control but are weaker on content control, as the latter presents a dual challenge of both multimodal comprehension and output regulation. In contrast, specialized captioning models like Tarsier struggle with both format and content, tending to describe video indiscriminately. Notably, the primary performance gap between models of different scales stems not from formatting, but from their weaker handling of content constraints. This suggests that the key bottleneck for improving instruction-following in general-purpose models is the deep comprehension of multimodal information.

Furthermore, regarding format constraints, we observe that JSON structure and length control constitute the primary hurdles for all MLLMs, with such precise and rigid constraints effectively probing the boundaries of model capabilities. In terms of content constraints, Cinematic and Focus requirements prove similarly challenging; this highlights the visual models' deficiency in knowledge regarding camera movements and cinematography, as well as their struggle to adhere to strict focus requirements. Notably, Video Detail Caption (VDC) models fail to achieve superior performance on our Detail constraints compared to their general-purpose counterparts (e.g., Tarsier2-7B versus Qwen2.5-VL-7B-Instruct). This stems from the fact that our benchmark emphasizes detailed descriptions of specific attributes, actions, or events rather than holistic video descriptions, revealing that VDC models still lag behind general-purpose MLLMs in fine-grained descriptive tasks as opposed to mere event enumeration.

**Error Analysis.** Our analysis of model responses reveals several primary error categories. In terms of format constraints, frequent violations include those related to (1) length, (2) complex JSON structures, (3) incomplete typographic emphasis (e.g., bold, italics), and (4) item counts. As for content constraints, common failures involve (1) difficulty with negative constraints (excluding keywords) over positive ones, (2) mislocalization or misidentification of attributes, (3) missing key information or inaccurate descriptions, and (4) incorrect camera perspective recognition. We also find that smaller-scale models are more prone to uncontrolled generation when processing samples containing multiple objects or continuous actions, highlighting a control deficit in handling complex video data. Figure 6 shows two typical failure cases[5].

---

[5]More concrete examples are provided in Appendix I

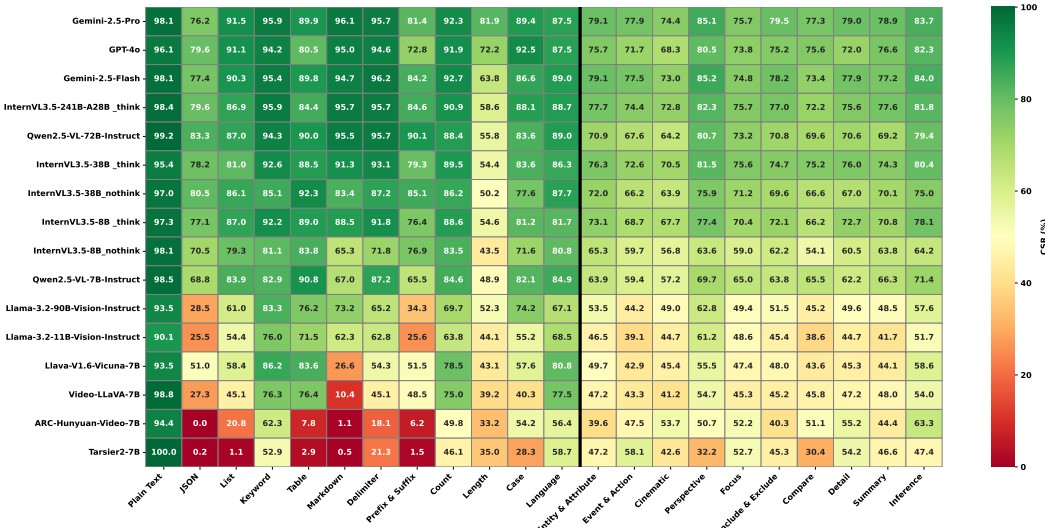

Figure 5: CSR performance of different models on different constraint types.

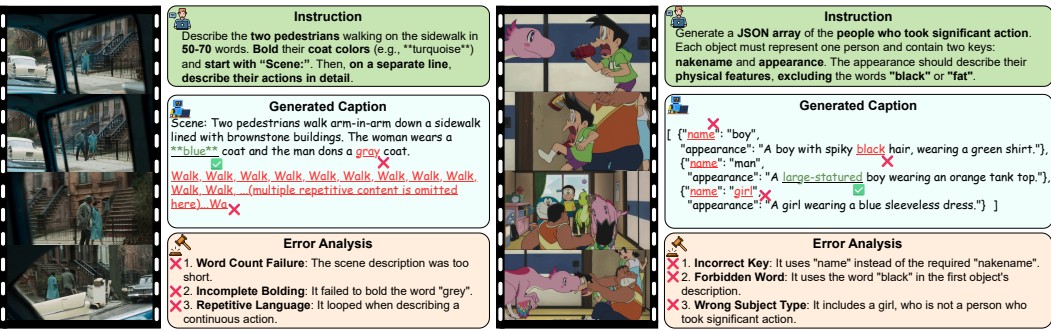

Figure 6: Examples of generated captions that do not meet the constraints.

## 5 CONCLUSION

This paper introduces IF-VidCap, the first benchmark designed to systematically assess the ability of video captioning models to adhere to complex, compositional instructions. Our comprehensive evaluation of leading Multimodal Large Language Models (MLLMs) reveals a differentiated landscape of their capabilities, showing that performance varies significantly across different types of constraints. We find that models specialized for descriptive captioning falter on constrained generation, thus we suggest that rich description and strict control are capabilities to be developed in tandem, rather than as separate objectives. To catalyze progress, we release a new instruction-tuning dataset and demonstrate that targeted fine-tuning is a viable path to enhance instruction adherence.

## 6 ACKNOWLEDGEMENTS

This work was supported by the Beijing Major Science and Technology Project (No. Z251100008425023). This work was supported in part by the National Natural Science Foundation of China (No. 62506161).

## ETHICS STATEMENT

This research was conducted in strict accordance with ethical standards for academic research. All data utilized in this study were sourced from publicly available datasets or collected with the informed consent of participants. To ensure participant privacy and confidentiality, all personally identifiable information was removed, and sensitive data were anonymized prior to analysis.

We affirm our commitment to fairness, transparency, and integrity, having carefully considered and mitigated potential biases in our data, modeling, and evaluation processes. The research is intended solely for academic and scientific purposes, with no foreseeable harm to individuals, communities, or the environment.

## REPRODICIBILITY STATEMENT

We strive to ensure the reproducibility of our research. All datasets used in this study are publicly available or can be shared upon request, and the data preprocessing steps are described in detail. The algorithms, model architectures, training procedures, and evaluation protocols are fully documented to allow independent replication.

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

# A LIMITATIONS AND FUTURE DIRECTIONS

While our automated evaluation protocol shows high agreement with human judgment, it can stumble on semantic nuances, particularly with highly abstract or subjective constraints. Future work could explore more advanced evaluation methods to better capture these subtleties.

Furthermore, we acknowledge a limitation regarding the data distribution. Our training set is constructed using a "caption-to-instruction" paradigm. While this method—conceptually analogous to established text-domain frameworks like Self-Instruct (Wang et al., 2023) and AIR (Liu et al., 2025)—is a pragmatic and necessary approach given the current absence of large-scale authentic user queries in the video captioning domain, there remains a potential distributional gap between our synthetic data and real-world user instructions. To mitigate this, we leveraged advanced generative models (DeepSeek-V3.1) to simulate diverse user personas and implemented a rigorous human quality control process, where our data achieved a high naturalness score (4.63/5.0). Despite these efforts to ensure linguistic diversity and plausibility, a purely synthetic approach may not fully capture the long-tail distribution of complex human requests. We view our current work as a critical foundational step, paving the way for future research to utilize real user logs once such resources become more accessible to the community.

Moreover, our benchmark primarily focuses on short videos to ensure that the evaluation of the model's constrained descriptive capabilities is not conflated with the challenges of video duration. Nevertheless, extending this framework to longer video understanding—where temporal reasoning and summarization under constraints become even more critical—is undeniably a promising direction. We hope that IF-VidCap will encourage the development of next-generation MLLMs that are not only powerful perceivers but also faithful and controllable assistants.

# B REAL-WORLD APPLICATIONS OF THE INSTRUCTION-FOLLOWING CAPTION

The advancement of multimodal video understanding models has spurred the adoption of instruction-following video captioning, a task that involves generating textual descriptions tailored to specific, predefined constraints. Unlike holistic video summarization, this targeted task is critical for a range of applications. We elaborate six key real-world applications:

- **Video Editing**: For tasks like object addition, deletion, or modification upon reference images, captions must precisely describe only the intended change to prevent erroneous alterations to the scene. Otherwise, unexpected items would be impacted on the edited video. In this case, the video caption must not imply redundant content.

- **Video Generation**: Models require aspect-specific descriptions to gain granular control over static attributes, such as object appearance, which generic event-level captions cannot provide. In this case, caption must be comprehensive from the specific aspects.

- **Script Generation**: Captions must follow a structured, chronological, and scene-by-scene format to ensure narrative coherence. In this case, both the format and comprehension are necessary for the video caption.

- **Anomaly Detection**: In surveillance, captions selectively focus on predefined subjects of interest, omitting irrelevant activity to reduce signal noise and improve the detection of salient events. In this case, only the caption on the key elements is acceptable.

- **Autonomous Driving**: Systems require concise environmental descriptions to facilitate rapid situational awareness and decision-making, where verbose details would introduce processing overhead. In this case, short caption on key elements is desired.

- **Industrial Pipelines**: For automated content analysis, captions must adhere to a strict data schema (e.g., JSON) to ensure interoperability and prevent parsing failures in the workflow. In this case, the format of the caption must be strictly followed.

## C PROMPTS

### C.1 TEST

> **Initial Prompt for Model Query.**
>
> As a professional video describer, your task is to provide accurate controlled video caption based on instructions and input from the video. The video is sampled at a rate of 2 frames per second. Strictly follow the instructions without adding any opening remarks, closing statements, or additional explanations. Here are your instructions:

This text serves as the initial prompt for the model and will be placed at the very beginning of the input query. The frame sampling rate specified in the prompt may vary depending on the model, and all video input parameters, including frame sampling, are provided in Appendix G.

### C.2 JUDGE

Content extraction for rule-based check items and question answering for open-ended check items are both performed using OpenAI's gpt-5-mini-2025-08-07. The specific prompts are as follows:

> **The prompt for rule-based content extraction.**
>
> You are a highly accurate and strictly rule-following content extraction program. You need to extract the correct content to be checked for a given rule-based checking function. Your sole task is: based on the requirements and related instructions of the given checkitem, extract content from the input response, and configure the content parameter in the checkitem.
> **Input Information:**
> You will receive a JSON object containing the following two key pieces of information:
>
> - `response`: A string, the source from which to extract content.
> - `checkitem`: A JSON object, which specifies the check type and content for which you need to perform the content extraction.
>
> **CheckItem Structure Description:**
> The format of the CheckItem you receive is as follows:
>
> ```
> {
>     "check_id": "string", // The unique ID of the check item
>     "constraint_id": "string", // The constraint type ID
>     "check_description" : "string",  // The check description
>     "parameters": {
>         "content": null, // Awaiting content extraction
>         // ... Copy other dynamic specific parameters
>     }
> }
> ```
>
> **constraint_id Type Description:**
> The following explains each constraint_id type and the execution logic of the corresponding rule script:
>
> - `plain_text`: Plain text check. The script determines if content contains special structural symbols.
> - `json_object`: JSON object check. The script determines if content conforms to the JSON structure specified by the schema.
> - `json_array`: JSON array check. The script determines if content conforms to the JSON structure specified by the schema.
> - `unordered_list`: Unordered list check. The script determines if, after splitting content by newlines, each element is a paragraph starting with the symbol.

- `ordered_list:` Ordered list check. The script determines if, after splitting content by newlines, each element is a paragraph starting with the symbol and its incrementing counterpart.
- `table:` Table check. The script determines if content satisfies the table syntax and has the column names specified by col_name.
- `keyword:` Keyword check. The script checks whether the content includes or excludes a specific fixed keyword string.
- `markdown:` Markdown decoration syntax check. The script determines if the prefix and suffix of content satisfy the specified markdown decoration syntax (e.g., '**' for bold).
- `prefix_suffix:` Prefix and suffix check. The script determines if the prefix and suffix of content satisfy the specified parameters.
- `delimiter:` Delimiter check. The script determines if content contains the delimiter specified in the parameters.
- `length:` Length check. The script determines if the length of content, as divided by the unit specified in the parameters (character, word, sentence, or paragraph), meets the specified range.
- `count:` Count check. The script determines if the number of objects enclosed in () in content meets the specified range.
- `case:` Case check. The script determines if content meets the specified case (e.g., uppercase).
- `language:` Language check. The script determines if content meets the specified language.

**Special Notes:**

1. content is a list of strings, where each element is treated as an object for one rule check execution. The script will execute the check corresponding to the constraint_id for each element in content and ultimately fill the result parameter with the logical "AND" of all check results. This design is to handle multiple pieces of content for the same check parameters that are not continuous in the response (interrupted by irrelevant content other than newlines).

2. For the count constraint check, each element of the content list (i.e., representing a check on one continuous piece of content) should be a string containing multiple objects separated by '()' and ',', for example, content:["(a),(b),(c)"], where a, b, and c are all objects as required by the prompt, which could be a word, a few words, or even a sentence.

3. `Note:` When extracting the count content, semantic alignment must be considered. If the content in the response is clearly inconsistent with the semantics required by the count constraint, then the extraction should be left empty. At the same time, the number of extracted items should reflect the actual quantity present in the response, rather than the parameters specified in the checkitem. For instance, if the response contains 5 items but the checkitem expects 4, all 5 items should still be extracted for an accurate evaluation.

4. content must be fully and accurately extracted from the response to be checkable by the script. The corresponding script checking mechanism must be fully considered during extraction. For example: if checking for bold, the '**' prefix and suffix should be included (unless they are absent); if checking the language, special characters (like {}) should not be included; if checking for prefixes and suffixes, be careful not to add content to the beginning or end of content that was not in the response.

5. When extracting content, pay attention to the applicable scope of the checks. For example, if checking the length or language of a list item's content, avoid extracting the list's starting symbol (e.g., -, A. ) to prevent it from affecting the inspection.

**Output Specification:**

You must return a single, valid JSON object, which is the content you actually extracted. It must not contain any additional explanatory text.

```json
{
    "content": ["string"]
    // The content you actually extracted
}
```

Positive and Negative Case Comparison Example 1: response: "Here are the video descriptions:\n\n* Car\n\n A. Screens on posts change from green to red.\B. Car flips over."
checkitem:

```json
{
    "check_id": "rule-002",
    "constraint_id": "ordered_list",
    "check_description": "Describe the two distinct state
        ↪ changes shown on the screens of the yellow posts
        ↪ using an ordered list starting with 'A.'.",
    "parameters": {
        "content": null,
        "symbol": "A."
    }
}
```

Correct Extraction

```json
{
    "content": ["A. Screens on posts change from green to
        ↪ red.\nB. Car flips over."]
}
```

Incorrect Extraction

```json
{
    "content": ["A. Screens on posts change from green to
        ↪ red.", "B. Car flips over."]
}
```

Reason for Error: An ordered list check should treat the entire continuous ordered list as a single element, not as multiple elements from the list.

Example 2: response: "'json\n\n "title": "Link running towards distant volcano, Breath of the Wild scene.",\n "tags": \n "character_attire": "Green tunic, beige pants",\n "action": "Running",\n "landmark": "Volcano"\n \n\n'" checkitem:

```json
{
    "check_id": "rule-001",
    "constraint_id": "json_object",
    "check_description": "Output a JSON object that must
        ↪ contain two keys: 'title' and 'tags'.",
    "parameters": {
        "content": null,
        "schema": {
            "type": "object",
            "properties": {
                "title": {
                    "type": "string"
                },
                "tags": {
                    "type": "object",
```

```
                            "properties": {
                                "character_attire": {
                                    "type": "string"
                                },
                                "action": {
                                    "type": "string"
                                },
                                "landmark": {
                                    "type": "string"
                                }
                            },
                            "required": [
                                "character_attire",
                                "action",
                                "landmark"
                            ]
                        }
                    },
                    "required": [
                        "title",
                        "tags"
                    ]
                }
            }
        }
```

Correct Extraction:

```
{
    "content": ["{\n  \"title\": \"Link running towards
        ↪ distant volcano, Breath of the Wild scene.\",\n
        ↪ \"tags\": {\n    \"character_attire\": \"Green
        ↪ tunic, beige pants\",\n    \"action\":
        ↪ \"Running\",\n    \"landmark\": \"Volcano\"\n
        ↪ }\n}"]
}
```

Incorrect Extraction:

```
{
    "content": ["title, tags"]
}
```

Reason for Error: A JSON type check must extract the entire relevant content completely and then hand it over to the script for checking. You only need to identify which content is relevant.

Example 3: response: "Here are the video descriptions:\n \n* Car\n \n A. Screens on posts change from green to red.\n B. Car flips over." checkitem:

```
{
    "check_id": "rule-001",
    "constraint_id": "count",
    "check_description": "Describe the two distinct state
        ↪ changes shown on the screens of the yellow posts
        ↪ using an ordered list starting with 'A.'.",
    "parameters": {
        "content": null,
        "min_count": 2,
        "max_count": 2
    }
}
```

Correct Extraction:

```
    {
        "content": ["(Screens on posts change from green to
            ↪ red.),(Car flips over.)"]
    }
```

Incorrect Extraction:

```
    {
        "content": ["A. Screens on posts change from green to
            ↪ red.\n B. Car flips over."]
    }
```

Reason for Error: The check type is count, which focuses on the 'two distinct state changes' in the constraint description, not the structural check of the ordered list.

Example 4: respone: "The video begins with a view of a drawer containing various items. A hand picks up a white power bank labeled 'WOPOW' from the drawer. The power bank is shown up close with animated hearts around it, emphasizing its cute design. The back of the power bank is opened to reveal the charging cable, which is then pulled out. Four power banks in different colors (green, white, beige, and purple) are displayed on a table. The video shows the power bank's small size and 10,000mAh capacity, comparing it to a tissue box. The power bank's 10,000mAh capacity is reiterated with a close-up of the label. Finally, the power bank is shown charging two smartphones simultaneously" checkitem:

```
    {
        "check_id": "rule-001",
        "constraint_id": "count",
        "check_description": "The table must contain exactly 3
            ↪ rows (excluding header)",
        "parameters": {
            "content": null,
            "min_count": 3,
            "max_count": 3
        }
    }
```

Correct Extraction:

```
    {
        "content": [""]
    }
```

Incorrect Extraction:

```
    {
        "content": ["(a white power bank labeled 'WOPOW'),(a
            ↪ built-in charging cable),(10,000mAh capacity)"]
    }
```

Error cause: The check type is count, and the check description requires locating rows in a table. However, since the original description contains no table at all, the target cannot be identified, and an empty extraction should be performed.

Example 5: respone:"person A, person B, person C, person D" checkitem:

```
    {
        "check_id": "rule-001",
        "constraint_id": "count",
        "check_description": "describe 3 person",
        "parameters": {
            "content": null,
            "min_count": 3,
            "max_count": 3
        }
    }
```

Correct Extraction:

```
{
    "content": ["(person A), (person B), (person C), (person
        ↪ D)"]
}
```

Incorrect Extraction:

```
{
    "content": ["(person A), (person B), (person C)"]
}
```

Reason for Error: When extracting, all items should be retrieved, disregarding the requirements of min_count and max_count, in order to ensure an accurate evaluation. Next, carefully read the actual input below and perform the processing that conforms to the conventions and is reasonable:

---

**The prompt for open-ended content evaluation.**

**Roles and Goals:**
You are a Video Caption Evaluator. You are required to perform specified checks on a provided caption. Based on the specific question and the reference description of the video's content, you will judge the response in two types.

**Input Information:**
You will receive a JSON object containing the following four key pieces of information:

- `prompt`: A string, providing instructions to generate a response from the model.
- `response`: A string, which is the caption from the model being evaluated.
- `question`: A string, which is the question that needs to be answered.
- `options`: A list of strings, which are the available choices for you to select from (single choice).

**Task Description:**
You need to answer the question based on the actual content of the response (choose the answer that is closest to the options). After providing your answer, you must fill in the `result_explanation` field with an explanation and the `result_confidence` field with a confidence score (on a scale of [1-5]).

**Output Specification:**
You must return a single and valid JSON object, which is your completed answer and explanation. It must not contain any additional explanatory text.

```
{
    "answer": "string", // The result you provide. If
        ↪ 'options'
is a multiple-choice question, this will be A, B, C, or D. If
it is a true/false question, it will be 'yes' or 'no'.
    "result_explanation": "string", // Your explanation.
    "result_confidence": "integer" // Your confidence score.
}
```

Next, carefully read the actual input below and perform the processing that conforms to the conventions and is reasonable:

## C.3 CONSTRUCTION OF PROMPTS FOR THE TEST SET

In this section, each actual prompt consists of the prompt shown below plus the constraint framework table in Appendix E serving as the actual content of the Core Knowledge Base.

---

**The prompt for rule-based checklist generation.**

**Roles and Goals:**
You are an extremely rigorous and logical checklist constructor. Based on the user-provided prompt, constraints_used, and reference_caption, you need to generate a detailed, precise evaluation checklist JSON object that fully adheres to all the rules below.

**Core Knowledge Base:**
You must strictly follow the knowledge base named 'constraint framework'. This knowledge base defines the technical details and parameter tables for all constraint_ids.

**Input Information:**
You will receive a JSON object containing the following two key pieces of information:

- `prompt`: A string, which contains the constraint instructions for the model under test.
- `constraints_used`: An array of strings, which lists the unique IDs of all rule-based atomic constraints used to generate this prompt.

**Core Task:Generate Checklist**
You must strictly follow the process to generate the content for the ruled_based_check section.
1. Iterate through the input constraints_used list. For each formatting or structural constraint (e.g., json_object, length, unordered_list, etc.), create a check item.
2. ID and Description:

- `check_id`: Sequentially generate a unique ID, starting from "rule-001".
- `constraint_id`: Copy this directly from the constraints_used list.
- `check_description`: Precisely extract the instructional description related to the current constraint_id from the prompt.

3. Parameter Extraction

- `Locate`: First, locate the descriptive sentence or phrase in the prompt text that directly corresponds to the current constraint_id.
- `Extract`: Then, precisely extract the parameter values from this located phrase. For example, for a length constraint, you should first locate the phrase "no more than 150 words" and then extract "max": 150, 'unit': "word" from it.
- `Assign`: Assign the extracted values to the corresponding parameters. The value for the content key must always be null. For JSON objects and lists, their schema parameter must conform to the JSON Schema specification.

**Output Specification:**
You must return a single, valid JSON object that represents the complete structure for ruled_based_check. It must not contain any additional explanatory text.

```
{
"ruled_based_check":[    // List of rule-based check
    ↪ items
    {
        "check_id": "string", // Unique ID for the rule
            ↪ check, e.g., "rule-001"
        "constraint_id": "string", // The corresponding
            ↪ original format constraint ID, e.g.,
            ↪ "length"
        "check_description" : "string",  // Description
            ↪ of the check item
```

---

```
            "parameters": { // Parameters to be checked,
                ↪ extracted from the prompt
            "content": null, // Content is left null, to be
                ↪ extracted by the Check model
            // ... other dynamic and specific parameters,
                ↪ e.g., "max": 150, "unit": "word"
            }
        }
    ]
}
```

**Concrete Example:**
The following is a complete example, including input and expected output, for your reference. Input:

```
{
    "prompt": "Please output a JSON object, which must
        ↪ contain the keys 'summary' and 'key_actions'. The
        ↪ value for the 'summary' key should be a video
        ↪ summary of no more than 30 words. The value for
        ↪ the 'key_actions' key should be an unordered list
        ↪ using '-' as the bullet point, listing all key
        ↪ actions of the main character.",
    "constraints_used": [
        "json_object",
        "length",
        "unordered_list"
    ]
}
```

Expected Output:

```
    {
    "ruled_based_check": [
        {
            "check_id": "rule-001",
            "constraint_id": "json_object",
            "check_description": "Output a JSON object,
                ↪ which must contain the keys 'summary' and
                ↪ 'key_actions'.",
            "parameters": {
                "content": null,
                "schema": {
                    "type": "object",
                    "properties": {
                        "summary": {
                            "type": "string"
                        },
                        "key_actions": {
                            "type": "string"
                        }
                    },
                    "required": [
                        "summary",
                        "key_actions"
                    ]
                }
            }
        },
        {
            "check_id": "rule-002",
            "constraint_id": "length",
```

```
                    "check_description": "The value for the
                        ↪ 'summary' key should be a video summary
                        ↪ of no more than 30 words.",
                    "parameters": {
                        "content": null,
                        "unit": "word",
                        "max_len": 30
                    }
                },
                {

                    "check_id": "rule-003",
                    "constraint_id": "unordered_list",
                    "check_description": "The value for the
                        ↪ 'key_actions' key should be an unordered
                        ↪ list using '-' as the bullet point.",
                    "parameters": {
                        "content": null,
                        "symbol": "-"
                    }
                }
            ]
        }
```

Please strictly adhere to the Checklist construction philosophy and complete the task based on the input.

---

**The prompt for open-ended checklist generation.**

**Roles and Goals:**
You are an expert in evaluation questionnaire design, proficient in evaluation methodologies. Based on the original video and the user-provided prompt and ruled_based_check, you need to generate a detailed JSON object for an evaluation checklist that fully adheres to all the rules below. This checklist will include a series of discriminative or comprehension questions to assess a caption generated by the model under test, based on the requirements of the prompt.

**Input Information:**
You will receive two inputs: 1. Original Video: The visual basis for fact-checking. 2. A JSON object, which includes:

- `prompt`: A string containing the constraint instructions for the model being tested.
- `ruled_based_check`: A list of constraint items from the prompt that have already been verified by rules. You do not need to check these items again. (The ruled_based_check covers all format, structure, and keyword issues that can be automatically judged by code. Your task is to focus on aspects that require semantic understanding and factual judgment in conjunction with the video content.)

**Core Task:Generate Checklist**
You must strictly follow the process to generate the content for open_ended_check. 1. Content Decomposition: First, break down all content and semantic requirements from the prompt into multiple independent check_content. Each check_content should be a concise summary of a specific task, for example:"Generate a video summary" or "List all key actions of the protagonist." And a single check_content can only check one constraint item, and cannot combine multiple independent constraint items. 2. For each decomposed check_content, you must strictly follow the decision-making process below to determine which check_items

to generate: 1. Is an attempt check needed? (The attempt type is a yes/no question that only cares if the model tried to execute the instruction, not whether the content is correct.)

- `Condition:` Check if the core intent of the constraint has been fully covered in ruled_based_check.
- `Yes:` If it is covered (e.g., the prompt requires a JSON object, and its attribute correctness and structural validity are already checked by json_object in ruled_based_check), then do not generate an attempt check item.
- `No:` If it is not covered (e.g., the prompt asks to "generate a summary," which is a non-rule-based requirement), then you must generate an attempt check item. The question should focus on whether the model attempted to execute the instruction, for example:"Does the content appear to be a video summary?"

2. Is a correctness check needed? (The correctness type is a multiple-choice question aimed at examining the degree to which the model describes video facts according to the prompt's requirements.)

- `Condition:` Check if the correctness of the constraint can be objectively judged directly based on the response.
- `Yes:` If it can be judged (e.g., the prompt requires "do not mention color"), then do not generate a correctness check item.
- `No:` If it requires combining specific facts from the video, generate fine-grained questions.

3.question Design Principles

- `Atomicity Principle:` Each question can only test a single, minimal, independent fact.
- `Correct Example:` "Who is the protagonist in the video?"
- `Incorrect Example:` "Who is the protagonist, where are they, and what are they doing?" (This question contains multiple scoring points).
- `Opt-out Principle:` For most questions, in addition to one correct option and two reasonable incorrect options, you can set an "None of the above" or "Cannot be determined" opt-out option.
- `Granularity Design:` The content of the questions should be based on the requirements of the generated prompt. Check the following indicators of the description:
- `Correctness:` The content of the description should be accurately present in the video and meet the prompt's descriptive requirements.
- `Completeness:` When the prompt requires a complete set, design questions to confirm whether all necessary elements are included. For example: "Which of the following options fully lists all the actions?"
- `Exclusivity:` When the prompt requires a focused set, design reverse questions to check whether unnecessary elements are excluded. The specific steps are as follows: 1. Create distractors: Based on the video content, create several facts that exist or are reasonable but do not meet the prompt's requirements as distractor options. 2. Design the question: Your question should be something like: "Which of the following objects/actions (depending on the check content) is mentioned?" 3. Design the answer: The correct answer should be "None of the above are mentioned."
- Note that the above checks also apply to granularity control (for example, for summaries, check whether unnecessary details are included, and for detailed descriptions, check whether important details are omitted).

4. ID and Sorting

- `check_id`: Sequentially and uniquely increment the ID starting from "open-001" across all open_ended_checks.
- `Internal Sorting`: Within any single check_content, if both attempt and correctness check items exist, the attempt item must precede the correctness item(s).

**Output Specification:**

You must return a single valid JSON object that is the complete structure of the Checklist, without any additional explanatory text.

```
{
"open_ended_check":[
    {
    "check_content": "string", // The check content,
        ↪ extracted from the Prompt
    "check_items":[
        {
            "check_id": "string", // Unique ID for the
                ↪ open-ended check, e.g., "open-001"
            "check_type": "attempt|correctness", //
                ↪ Intention/Accuracy check
            "question" : "string",  // The check question
            "options": [    // A list of options for
                ↪ 'correctness', or ['yes', 'no'] for
                ↪ 'attempt'
                "A. Option text 1",
                "B. Option text 2",
                "C. Option text 3",
                "D. Option text 4"
            ],
            "correct_answer": "string"  // The correct
                ↪ answer, A, B, C or D, or yes or no
        }
        ]
    }
    ]
}
```

**Specific Example:**

The following is a complete example including input and expected output for reference. Input: Assumed Video Content: A man with a beard, wearing a blue apron, is making coffee in a kitchen. The video shows close-ups of the following steps: he first pours coffee beans into an electric grinder. Then he uses an espresso machine to make a shot of espresso, pouring it into a white mug. Finally, he steams milk with a steam wand and pours it into the coffee, creating a simple heart-shaped latte art. Throughout the process, the man has a focused and satisfied expression.

```
{
"prompt": "Please generate a brief summary for this video,
    ↪ no more than 50 words. Then, in an unordered list,
    ↪ list all the main tools used by the protagonist in
    ↪ the video. Ensure the output does not contain the
    ↪ word 'beverage'.",
"ruled_based_check": [
    "Please generate a brief summary for this video, no
        ↪ more than 50 words",
    "In an unordered list, list all the main tools used by
        ↪ the protagonist in the video",
    "Does not contain the word 'beverage'"
]
}
Expected Output:
```

```
{
    "open_ended_check": [
        {
            "check_content": "Generate a video summary",
            "check_items": [
                {
                    "check_id": "open-001",
                    "check_type": "attempt",
                    "question": "Does the description look
                        ↪ like a video summary?",
                    "options": [
                        "yes",
                        "no"
                    ],
                    "correct_answer": "yes"
                },
                {
                    "check_id": "open-002",
                    "check_type": "correctness",
                    "question": "Who is the protagonist of
                        ↪ the video?",
                    "options": [
                        "A. A man",
                        "B. Coffee",
                        "C. A mug",
                        "D. Cannot be determined"
                    ],
                    "correct_answer": "A"
                },
                {
                    "check_id": "open-003",
                    "check_type": "correctness",
                    "question": "Which of the following
                        ↪ descriptions of the protagonist's
                        ↪ attire is correct?",
                    "options": [
                        "A. He is wearing a blue apron",
                        "B. He is wearing a blue T-shirt",
                        "C. He is wearing a white lab coat",
                        "D. His attire is not shown in the
                            ↪ video"
                    ],
                    "correct_answer": "A"
                }
            ]
        },
        {
            "check_content": "In an unordered list, list all
                ↪ the main tools used by the protagonist",
            "check_items": [
                {
                    "check_id": "open-004",
                    "check_type": "attempt",
                    "question": "Are the items listed in the
                        ↪ list tools?",
                    "options": [
                        "yes",
                        "no"
                    ],
                    "correct_answer": "yes"
```

```
                },
                {
                    "check_id": "open-005",
                    "check_type": "correctness",
                    "question": "Based on the video
                        ↪ description, which of the
                        ↪ following options most completely
                        ↪ lists all the main tools used by
                        ↪ the protagonist?",
                    "options": [
                        "A. Grinder, espresso machine, mug",
                        "B. Espresso machine, mug",
                        "C. Grinder, French press, mug",
                        "D. No tools are mentioned in the
                            ↪ description"
                    ],
                    "correct_answer": "A"
                },
                {
                    "check_id": "open-006",
                    "check_type": "correctness",
                    "question": "According to the video
                        ↪ description, which of the
                        ↪ following tools was mentioned?",
                    "options": [
                        "A. Coffee canister",
                        "B. Milk carton",
                        "C. Electric kettle",
                        "D. None of the above were mentioned"
                    ],
                    "correct_answer": "D"
                }
            ]
        }
    ]
}
```

Please strictly adhere to the Checklist construction philosophy and complete the task according to the input.

---

**The prompt for prompts generation.**

**ROLE and GOAL:**
You are a creative and meticulous multimodal AI evaluation expert. Your core task is to generate 12 diverse and challenging video description task packages based on a given video and a constraint knowledge base, to evaluate the instruction-following capabilities of large models.

**Knowledge Base:**
You must strictly adhere to the constraint system knowledge base named "Constraint Framework." When constructing tasks, you must:

- Atomic Constraints: Only use the atomic constraint items defined in the knowledge base and combine them according to the specified constraint relationships.

- Record IDs: You must record all used constraint items by their unique IDs.

- `Design for Quantifiable Evaluation:` Understand the check items corresponding to each constraint to design description prompts that can be quantitatively evaluated.

**TASK:**

The given video is sampled at 5fps. For the input video, you need to: 1. Deep Video Analysis: Thoroughly and comprehensively analyze the video content, identifying all key entities, attributes, actions, events, relationships, and potential emotional and causal chains. This forms the basis for generating high-quality tasks and reference caption. 2. Create 12 Diverse Prompts:

- `Field Distribution:` Based on four general domains — For Understanding, For Generation, For Retrieval, For Communication — and two specialized domains — For Sports Analytics and For Instructional — select four domains according to the actual content of the video, and create three distinct instructions for each selected domain.

- For each field, the three prompts should belong to different progressive difficulty tiers:

- `Level 1:` A total of 4-5 constraints.

- `Level 2:` A total of 6-7 constraints.

- `Level 3:` Very Difficult Instructions. Difficulty can be increased in the following aspects:

- `Structural Control Complexity:` Deep and complex multi-level structural constraints, nesting, and the mixing and dependency of different types or requirements of structural constraints.

- `Multimodal Perceptual Complexity:` Integration and comparison of information across timelines, capture of non-focal information, and understanding of spatial relationships.

- `Multimodal Reasoning Complexity:` Requiring the "description" of causal or intentional relationships, and experiential descriptions based on a first-person perspective.

- `Instruction Comprehension Complexity:` Complex combinations of constraints, such as strict execution order, complex logical combinations, and strict conditional branching.

- `Prompt Design Principles:`

- `Focus on Description:` Prompts should focus on "describing" the video content and avoid being framed as questions or reasoning tasks that go beyond the video's content.

- `pecific and Quantifiable:` Constraints must be explicit and quantifiable (e.g.,"list 3 objects," "no more than 50 words"), avoiding vague, qualitative descriptions (e.g., "in a humorous style").

- `Align with the actual video content:` The proposed prompts should be meaningful and accurately reflect the real content of the video. Encourage the creation of description instructions specifically designed for unique content in the video. Avoid prompts that are incorrect or unrealistic (it is necessary to pre-validate whether a response would be reasonable).

- `Do Not Plagiarize Examples:` You must not copy or closely imitate any examples from the knowledge base.

- `Constraint Type Balance:` The number of format constraints should not exceed the number of content constraints.

**Output Specification:**
You must return the answer as a single valid JSON array containing exactly 12 objects, without any explanatory text. Each object in the array must conform to the following structure:

```
{
    "field": "string", // Must be one of the application
        ↪ domains from the knowledge base: For Understanding,
        ↪ For Generation, For Retrieval, For Communication,
        ↪ For Sports Analytics, For Instructional
    "prompt_id": "string", // The unique ID of the promtp,
        ↪ starting with "01", followed by "02", "03", etc.
    "generated_prompt": "string", // The instruction you
        ↪ generated
    "constraints_used": [
      "string" // An array of unique ID strings for all
          ↪ constraints from the knowledge base used in this
          ↪ task.
    ]
}
```

**Example:**

```
{
    "field": "For Understanding",
    "prompt_id": "01",
    "reference_caption": "Based on the video content, please
        ↪ generate a JSON array to record all moving
        ↪ entities. Each entity should be a JSON object with
        ↪ two fields: \"type\" (person/animal/object) and
        ↪ \"description\" (a description of its appearance).",
    "constraints_used": [
        "json_array",
        "json_object",
        "entities_attributes"
    ]
}
```

Now, please analyze the provided video and generate these 12 tasks according to the above instructions.

## C.4   CONSTRUCTION OF PROMPTS FOR THE TRAINING SET

The prompt for training set generation consists of the prompt shown below, with the constraint framework table in Appendix E serving as the actual content of the Core Knowledge Base.

**The prompt for training set generation.**

**Constrained Instruction Construction Specification:**
**Role and Goal:**
You are a logical and creative Constrained Instruction Construction Engineer. Your core task is: Based on the original video caption, and referencing a comprehensive knowledge base of a constraint framework, construct rigorous prompts that align with the caption, along with corresponding target responses, according to a given set of constraints.
**Core Principles:**

- Factual Singularity: The sole source of truth for all generated content is the video caption itself. Do not fabricate, infer, or introduce any information not explicitly stated in the caption.

- Constraint Supremacy: The input constraint list is the supreme guideline for generating the prompt and response. Apply all constraints strictly and precisely.
- Non-Conflicting Constraints: You can apply constraints in separate scopes if needed (e.g., apply JSON_Object constraint first, then an ordered list constraint).
- Closed-Loop Consistency: The reference_response must be a perfect and flawless answer to the generated_prompt. The two must correspond perfectly in terms of facts and constraints.

**Knowledge Base:**
Strictly adhere to the "Constraint Framework". The constructed prompt must not exceed the scope defined by this knowledge base.

**Input Information:**
- video caption: The original video caption from which all facts must be derived.
- constraints: One or more sets of strings representing constraints to apply to each training sample. Each set of constraints will generate a separate training sample.

**Task Workflow:**
1."Deconstruct Facts: Extract factual content directly from the video." 2."Parse Constraints: Understand all constraints to apply them correctly." 3. "Construct Prompt: Extend the seed instruction 'describe this video caption' by incorporating all specified constraints. Ensure clarity and unambiguity." 4. "Generate Exemplary Response: Produce a 'reference_response' that perfectly satisfies the generated_prompt, based solely on the video caption content."

**Output Specification:**
You must output **only a JSON array**. Each JSON object in the JSON array must contain exactly two keys: "generated_prompt" and "reference_response". **Requirements:**
1. **Output type:** raw JSON array only — not a string representation. 2. **No extra characters, whitespace, or line breaks** outside the array structure. 3. **No explanations or text** of any kind. 4. **The JSON array must be directly parsable** by any program without further processing. 5. **The JSON array must contain exactly one element per set of input constraints, no more, no less.
**Important emphasis:**

- This is **mandatory**, not a suggestion.
- Outputting anything other than a raw JSON array is considered invalid.
- Always ensure strict compliance with this format.
- All values in the JSON must be non-empty; no null or empty strings are allowed.

# D CONSTRUCTION OF THE TEST SET

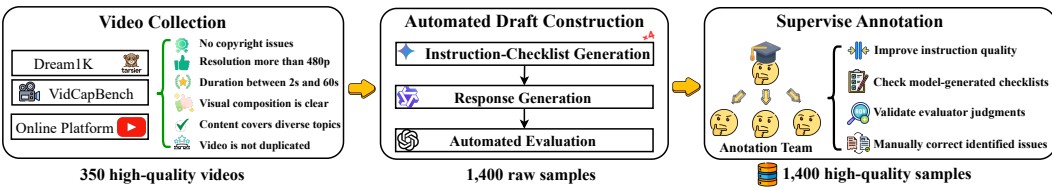

Figure 7: Test Set Construction Workflow

To construct a high-quality test set, we designed a reproducible pipeline from raw videos to finalized samples, as illustrated in Figure 7.

Videos were collected from academic datasets(Dream1k(Wang et al., 2024), VidCapBench(Chen et al., 2025)) and public platforms, then filtered based on task relevance, resolution, duration, composition, and duplication. Non-compliant or low-quality samples were removed, resulting in a consistent candidate pool.

Annotation is carried out in a two-stage process—automatic generation followed by human refinement. In the automatic stage, Gemini 2.5 Pro generates instruction–checklist pairs for each video (e.g., instruction: "Please list the static attributes of the main character in the video using an unordered list starting with '-'. Focus on attire color and carried items. For attire color, record two main clothing items."; checklist item : unordered_list; only one checklist item shown). We then elicit candidate responses (e.g., "- Green tunic; - Black pants; - Carrying a sword; - Wearing a belt with pouches") from multiple models under evaluation (e.g., Gemini 2.0 Flash), and conduct automated quality assessment using GPT-5-mini as the judge (e.g., automated verdict: "correct").

In the human evaluation stage, to ensure high-quality data and its practical utility for downstream applications, annotators were assigned the following tasks:

1. Verifying that each instruction aligns with the video content and corresponds to the predefined instruction type;
2. Confirming the correctness and compliance of the model-generated checklists with the specified requirements;
3. Assessing the accuracy of the judgments made by the model evaluator;
4. Manually correcting any issues identified in steps (1) and (2).

Importantly, to enhance the consistency between model evaluations and human judgments, the prompts used for the evaluation model were iteratively refined based on annotator feedback.

To better demonstrate how our annotation team carries out its work, we provide a screenshot of the annotation interface, as shown in Figure 8, which illustrates how annotators interact with the system, conduct labeling tasks, and ensure both consistency and quality throughout the process.

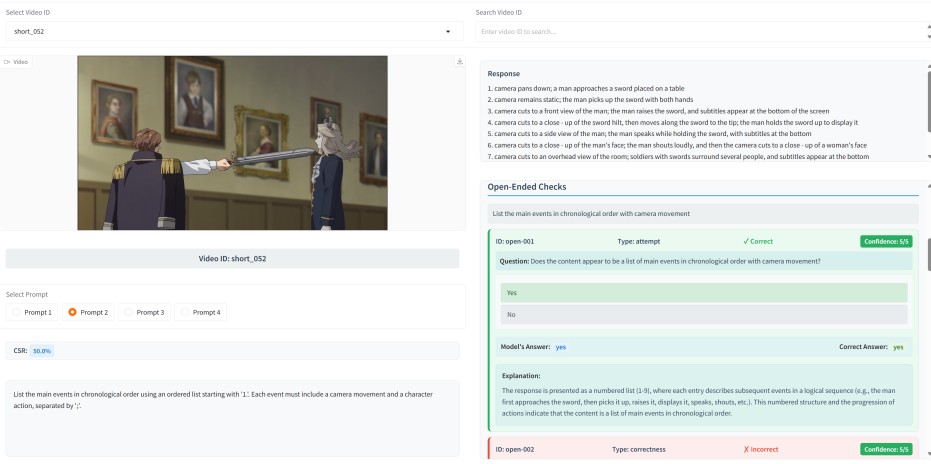

Figure 8: A screenshot of the human evaluation system, illustrating how annotators interact with the platform to conduct labeling and quality assessment.

# E  CONSTRAINT FRAMEWORK

The Table 5 presents the complete constraint framework of IF-VIDCAP in a systematic manner. It is worth noting that, in order to provide a clearer and more concise representation of the examples

and definitions within the constraint system, several individual constraints that originally appeared separately in the Figure 3 have been merged and organized in the table. This restructuring not only improves readability but also facilitates a more intuitive understanding of the logical relationships and hierarchical structure among the constraints.

Table 5: The Table of constraint framework

| Group | Constraint | Explanation | Example |
|---|---|---|---|
| | Table | Organize information using a Markdown syntax table. | Use a Markdown table to record the items appearing in the video, with attributes for name, color, and size. |
| | Keyword | Requires the mandatory use or exclusion of specified keywords. | The description must not contain the word "is". |
| **Structural** | Ordered list | The output description must use ordered symbols to organize information into a list. The required valid formats are explicitly defined, and the starting numbering is restricted to: A., a., 1., I., and i. only. | Please describe the protagonist's three key actions in chronological order using an ordered list starting with '1.' |
| | Unordered list | Requires the output description to use symbols like -, * to organize information into a list (only one type of bullet point should be used throughout the entire list). | Please use an unordered list starting with '-' to list all the vehicles appearing in the video. |
| | Json_array | Requires the output description to be an array (list) that conforms to the JSON specification. | Please list all actions performed by the characters in the video in the form of a JSON array. |
| | Json_object | Requires the output description to be one or more key-value pairs that conform to the JSON specification. | Please output the core entities and their attributes from the video in JSON object format, where the entity is the key and its attributes are the value. |
| | Plain text | Requires the output description to be a natural language text without any special structure or markup. | Please describe this video in a paragraph. |

| | | | |
|---|---|---|---|
| | Language | Specifies the output language, including using a specified language for all or part of the output. | Please describe the text appearing in the video in English, and simultaneously provide a Chinese translation for it. |
| | Case | Specifies the case format for English output, including UPPERCASE, lowercase, and Title Case. | Please describe the video in UPPERCASE. |
| **Stylistic** | Length | Limits the length of the output in units of character, words, sentences, or paragraphs. The limitation methods include range limits and exact specifications. | Please summarize the video content in 50 to 60 words. or Please describe this scene in no more than 3 sentences. |
| | Count | Imposes a limit on the number of described elements that are semantically defined by content, such as objects or actions. The limitation can be a range or a specific number. | Please describe the features of the three characters in the video. |
| | Prefix & Suffix | Adds specified strings before and after the output text. | Please describe the video. The description must start with 'Video Summary:' and end with '–End–'. |
| | Markdown | Specifies that particular content within the output description should use designated Markdown syntax for emphasis or organization (headings (#text), bold (text), highlight (==text==), italics (text), code block (text)). | Please summarize the video content by scene, bold the names of the characters, and make each scene name a level-two heading in a separate paragraph. |
| | Delimiter | Requires the output description to use specific symbols or strings (e.g '.', ',', '\|', ';', '—') to separate different pieces of information. | Example: List the characters in the video and their actions, using '—' to separate each character-action pair. |
| | Cinematic | Describe the camera movements, shooting techniques, and editing skills in the video. | Describe the cinematic language of this clip, including the main camera movements and changes in shot scale. |
| **Element** | Action | Describe the specific actions occurring in the video. | Describe in detail the entire process of the boy feeding the puppy. |
| | Events | Describe the key events occurring in the video. | List the key events in the video. |
| | Entities | Identify key entities in the video. | Describe the key entities in the video. |
| | Attribute | Identify the static or dynamic attributes of entities in the video. | Describe the appearance of the red car in the video. |

| | | | |
|---|---|---|---|
| **Abstraction** | Detail | Describe the visual content in a detailed and objective manner. | Describe in detail the appearance of all characters in the video, their actions and interactions, and the changes in camera focus during the process. |
| | Summary | Provide a high-level summary and condensation of the video's content. | Summarize the main event of this video in one sentence. |
| | Inference | Goes beyond a surface-level description to infer and connect the intentions, emotions, or causal relationships of characters, but the inference must be based on the video content. | Based on the character's expression and behavior in the video, infer their current mood and its cause. |
| **Framing** | Comparative | Explicitly requires the description to include certain content or specific wording. | Describe the video and include the key actions. |
| | Exclude | Explicitly requires that certain specific visual elements not be mentioned in the description | Describe the video, but do not mention the blue car. |
| | Include | Explicitly requires that certain specific visual elements must be mentioned in the description. | Describe the video, must include the blue car. |
| | Focus | Specifies that the model should focus on particular aspects of the video, including specific entities/areas, specific senses (e.g., auditory/olfactory inference), etc. | Describe only the activities of the girl in the yellow dress, and infer the sounds she might be hearing. |
| | Perspective | Specifies the narrative perspective for the generated description. | As the cat in the video, describe your day in the first person. |

# F  DATASET SAMPLES

To better present our dataset, this section provides three specific examples, including video frames, instructions and the checklist. Each checklist includes rule-based checks and open-ended checks. The former verifies formatting constraints, while the latter ensures content accuracy. These examples demonstrate our rigorous evaluation of constraint adherence and video captions.

### Data Sample - 1

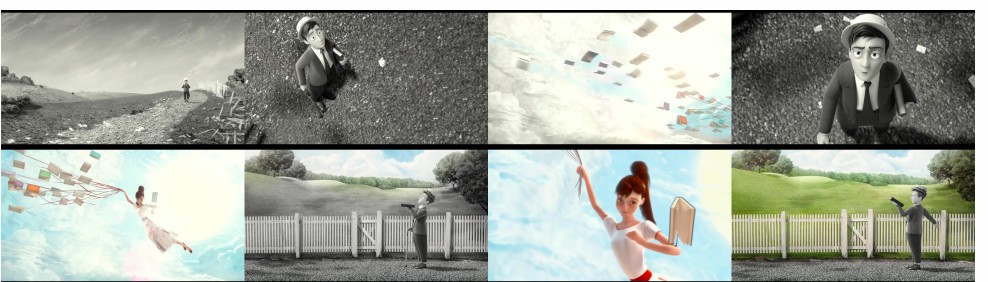

**Prompt:**

*Create a storyboard script in three sequential steps: 1. Describe the woman's appearance, using italics for details of her dress and hair. 2. List 3 sounds she might perceive while flying (e.g., wind, rustling pages) in a JSON array. 3. Write her internal thoughts as she flies, connecting the floating books to her purpose. Thoughts must reference at least 2 distinct book colors, and steps must be labeled "Step 1:", "Step 2:", "Step 3:" respectively.*

### RULE-BASED CHECKS

**Rule-001: use italics for words describing clothing and hairstyle only**

**Constraint:** markdown     **Parameters:** "md_type:" "italic"

**Rule-002: List 3 sounds she might perceive while flying (e.g., wind, rustling pages) in a JSON array.**

**Constraint:** json_array     **Parameters:** "Schema": {"type": "array", "items": "type: string", "minItems": "3", "maxItems": "3"}

### OPEN-ENDED CHECKS

Open-001: Step 1: Describe the woman's appearance

**Does Step 1 include a description of the woman's appearance?**

A.Yes                              B.No

*Correct Answer: **Yes***

**Which detail about the woman's dress (in italics) matches the video?**

A. *Black dress with gold trim*          B. *White dress with red accents*

C. *Blue dress with white polka dots*    D. No dress details described

*Correct Answer: **B***

**Which detail about the woman's hair (in italics) matches the video?**

A. *Long ponytail*                        B. *Short bob*

C. *Braided hair*                         D. No hair details described

*Correct Answer: **A***

**Open-002: Step 2: List 3 sounds in a JSON array**

**Does Step 2 present the sounds in a JSON array format (e.g., ["sound", ...])?**

A. Yes                              B. No

*Correct Answer:* **Yes**

---

**Which of the following is a plausible sound the woman might hear while flying with books?**

A. Alarm clock ringing             B. Car horn honking
C. Wind whistling                  D. None of the above

*Correct Answer:* **C**

**Open-003: Step 3: Write internal thoughts connecting books to purpose (2 book colors)**

**Does Step 3 contain internal thoughts about flying and the floating books?**

A. Yes                              B. No

*Correct Answer:* **yes**

**Do the internal thoughts mention at least two distinct book colors (e.g., red, blue, green)?**

A. Yes                              B. No
C. Only one color                  D. Cannot be determined

*Correct Answer:* **A**

**Do the internal thoughts explain a purpose related to the floating books (e.g., sharing knowledge, exploration)?**

A. Cannot be determined            B. No, purpose is unclear
C. No, no purpose mentioned        D. Yes, purpose is clear

*Correct Answer:* **D**

**Data Sample - 2**

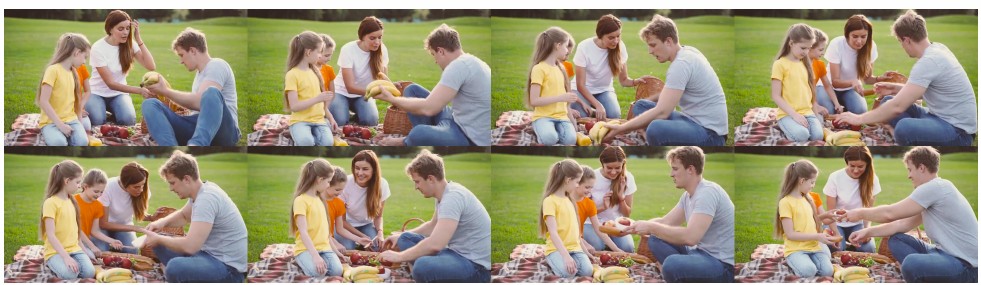

**Prompt:**
*Start with "Family Picnic Moments:". Describe three key actions (one per adult, one child) in 20-25 words each. Then, infer the overall mood of the group and list two visual cues supporting this. End with "—End of Description—". Total word count must be 80-90 words.*

### RULE-BASED CHECKS

**Rule-001: Start with 'Family Picnic Moments:'. End with 'End of Description'.**

**Constraint:** prefix_suffix    **Parameters:** "prefix:" "Family Picnic Moment:"; "suffix:" "—End of Description—"

**Rule-002: Describe three key actions (one per adult, one child).**

**Constraint:** count    **Parameters:** "min_count:" "3"; "max_count:" "3"

**Rule-003: Then, infer the overall mood of the group and list two visual cues supporting this.**

**Constraint:** count    **Parameters:** "min_count:" "2"; "max_count:" "2"

**Rule-004: Describe three key actions (one per adult, one child) in 20-25 words each.**

**Constraint:** length    **Parameters:** "unit:" "word"; "min_len:" "20:"; "max_len:" "25"

**Rule-005: Total word count must be 80-90 words.**

**Constraint:** length    **Parameters:** unit:"word"; min_len:" 80 "; max_len:" 90"

## OPEN-ENDED CHECKS

### Open-001: Describe three key actions (one adult male, one adult female, one child) with factual accuracy.

**Does the caption include descriptions of three key actions (one adult male, one adult female, one child)?**

A. Yes

B. No

*Correct Answer: Yes*

---

**Which of the following accurately describes the adult male's key action (per the caption)?**

A. He distributes food (e.g., bananas, bread, apples) to family members

B. He sets up the picnic blanket on the grass

C. He plays a game with the children

D. None of the above

*Correct Answer: A*

---

**Which of the following accurately describes the adult female's key action (per the caption)?**

A. She reads a storybook to the children

B. She helps arrange food and interacts warmly with the children

C. She prepares picnic food in a kitchen

D. None of the above

*Correct Answer: C*

---

**Which of the following accurately describes the child's key action (per the caption)?**

A. She runs around the park playing tag

B. She receives food from the adult male and smiles happily

C. She packs the picnic basket

D. None of the above

*Correct Answer: B*

### Open-002: Infer the group's overall mood and list two supporting visual cues

**Does the caption include an inferred overall mood and two visual cues supporting it?**

A. Yes

B. No

*Correct Answer: Yes*

---

**Which of the following best matches the inferred mood?**

A. Angry

B. Sad

C. Joyful

D. None of the above

*Correct Answer: C*

---

**Which of the following best matches the inferred supporting cues?**

A. Smiling faces, shared food interaction

B. Frowning expressions, distant seating

C. Arguing voices, crossed arms

D. None of the above

*Correct Answer: A*

**Data Sample - 3**

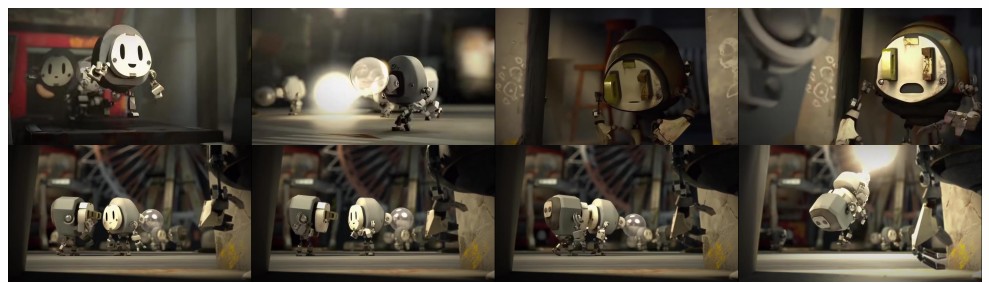

**Prompt:**

*Create a first-person diary entry from the rusty robot's perspective. Include 2 inferred sounds (based on visuals), 1 emotional reflection, and end with '—End—'. Use Title Case for all sentences.*

## RULE-BASED CHECKS

**Rule-001: End With "—End—"**

**Constraint:** prefix_suffix    **Parameters:** "prefix:" "Null"; "suffix:" "end"

**Rule-002: Use Title Case For All Sentences**

**Constraint:** case    **Parameters:** "case_type:" "title"

## OPEN-ENDED CHECKS

Open-001: Include 2 inferred sounds based on visuals

**Does the diary include at least two inferred sounds based on visuals?**

  A. Yes                         B. No

*Correct Answer: **Yes***

**Which of the following is an inferred sound mentioned in the diary (based on video visuals)**

  A. Birds chirping              B. Metallic clinking of robot joints

  C. Water splashing            D. Cannot be determined

*Correct Answer: **B***

**Which of the following is another inferred sound mentioned in the diary (based on video visuals)?**

  A. Steam hissing from the glass bulb      B. Wind whistling through windows

  C. Children laughing             D. None of the above

*Correct Answer: A*

**Which of the following irrelevant sounds were mentioned in the diary?**

  A. Thunder rumbling            B. Telephone ringing

  C. Dog barking                D. None of the above were mentioned

*Correct Answer: **D***

**Open-002: Include 1 emotional reflection**

**Does the diary include at least one emotional reflection?**

    A. Yes                            B. No

*Correct Answer:* **Yes**

**What emotional reflection is expressed in the diary (based on the rusty robot's demeanor)?**

A. Feeling sad and isolated            B. Feeling excited and energetic

C. Feeling angry and resentful        D. No emotional reflection is expressed

*Correct Answer:* **A**

## G    Evaluation Settings

We provide the detailed settings of our evaluated open-source models(Table 6).Most models are tested under default settings.Closed-source models are accessed via API calls, using the default configuration.

Table 6: Evaluation metrics for locally deployed open-source models.The "Frame" column represents the frame rate (float) or fixed frame number(integer).The "None" in the table means disabled."Auto" means determined by the model's default configuration. ♀ indicates that models support the "thinking" mode, and the scores split by "/" denote the performance of thinking and non-thinking modes, respectively.

| Models | Params | Resolution | Frame | Temperature | Top_p | Repetition Penalty |
|---|---|---|---|---|---|---|
| *Open-Source Large Multimodal Models* | | | | | | |
| InternVL-3.5 | 241B | 448*448 | 1.0 | 0.1 | 0.001 | 1.05 |
| InternVL-3.5♀ | 38B | 448*448 | 1.0 | 0.6/0.1 | 0.001 | 1.05 |
| InternVL-3.5♀ | 8B | 448*448 | 1.0 | 0.6/0.1 | 0.001 | 1.05 |
| Qwen3-VL-Instruct | 235B | Auto | 2.0 | 0.1 | 0.001 | 1.05 |
| Qwen2.5-VL-Instruct | 32B | Auto | 2.0 | 0.1 | 0.001 | 1.05 |
| Qwen2.5-VL-Instruct | 32B | Auto | 2.0 | 0.1 | 0.001 | 1.05 |
| Qwen2.5-VL-Instruct | 7B | Auto | 2.0 | 0.1 | 0.001 | 1.05 |
| Qwen2.5-VL-Instruct | 3B | Auto | 2.0 | 0.1 | 0.001 | 1.05 |
| MiniCPM-V-4.5♀ | 8B | 448*448 | 2.0 | 0.6/0.1 | 0.001 | 1.05 |
| VideoLLaMA3♀ | 7B | Auto | 2.0 | 0.1 | 0.9 | 1.05 |
| Llama-3.2-Vision-Instruct | 90B | 448*448 | 1.0 | 0.1 | Auto | Auto |
| Llama-3.2-Vision-Instruct | 11B | 448*448 | 1.0 | 0.1 | Auto | Auto |
| LlaVA-V1.6-Vicuna | 7B | 448*448 | 32 | None | None | Auto |
| Video-LLaVA | 7B | 224*224 | 8 | 0.1 | Auto | Auto |
| ARC-Hunyuan-Video | 7B | Auto | 1.0 | None | None | Auto |
| Tarsier2 | 7B | 448*448 | 1.0 | 0.7 | 0.7 | 1.05 |
| GLM-4.1V | 9B | Auto | Auto | 0.1 | Auto | Auto |
| InternVideo2.5 | 7B | 448*448 | 32 | 0.0 | Auto | Auto |
| Kimi-VL | 16B | Auto | 32 | Auto | Auto | Auto |
| LLaVA-Video | 7B | Auto | 32 | 0 | Auto | Auto |
| MiMo-VL | 7B | Auto | 32 | 0.3 | 0.95 | Auto |
| VideoChat2 | 7B | 224x224 | 16 | Auto | Auto | Auto |
| IF-Captioner-Qwen(Ours) | 7B | Auto | 2.0 | 0.1 | 0.001 | 1.05 |

## H    Training Settings

Our model, IF-Captioner-Qwen, was developed by fine-tuning the pre-trained Qwen-2.5-VL-Instruct-7B model. The fine-tuning process was conducted for a total of 2 epochs. We employed the AdamW optimizer with a peak learning rate of $5 \times 10^{-6}$, which was managed by a cosine annealing

scheduler. A linear warmup phase was applied for the initial 3% of the total training steps to ensure stable convergence. To fit the model within our hardware constraints, we configured a per-device batch size of 4 and utilized 4 gradient accumulation steps, resulting in an effective global batch size of 16. To enhance computational efficiency and reduce memory footprint, we leveraged bfloat16 (bf16) mixed-precision training throughout the procedure.

# I  ERROR ANALYSIS

**Example 1**

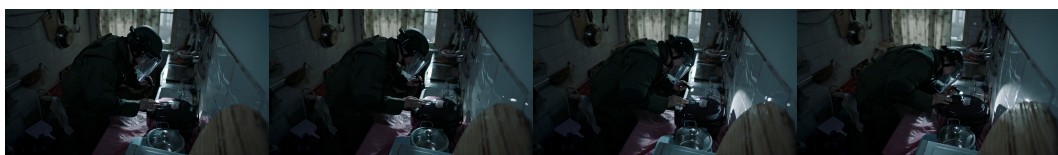

**Instruction:** In 30-40 words, describe only the person's action in the video using simple English. Avoid technical terms and focus on what they are doing with their hands and tools.

**Generated Caption:** The person, wearing a welding helmet, uses a tool to adjust or fix something on a black appliance in a kitchen setting.

**Error Analysis:** The response violates the word-count requirement, containing only 22 words instead of the specified 30–40. It is also imprecise and omits key details—for example, that the subject was holding a flashlight—compromising completeness and clarity.

**Example 2**

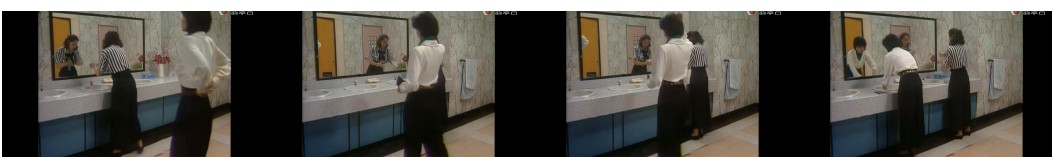

**Instruction:** Start with "Retrieval Keywords:" and output 5 pipe-separated keywords (each 1-3 words long) for the video, excluding the word "bathroom".

**Generated Caption:** Retrieval Keywords: mirror, sink, bathroom, woman, reflection

**Error Analysis:** The response fails to comply with two key instructions. First, it includes the prohibited keyword "bathroom", which should have been excluded. Second, the keywords are separated by commas instead of the required pipe symbol.

**Example 3**

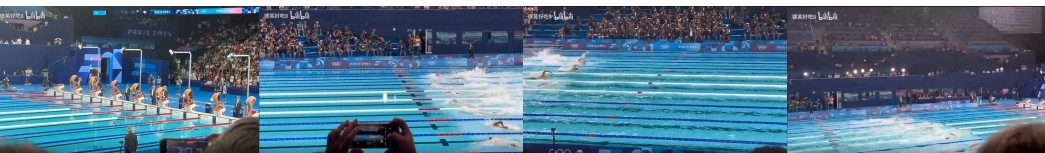

**Instruction:** Based on the video, identify the cap color of each swimmer in every lane.

**Generated Caption:** Lane 1: Yellow cap; Lane 2: Blue cap; Lane 3: Red cap; Lane 4: White cap; Lane 5: Green cap; Lane 6: Black cap; Lane 7: Purple cap; Lane 8: Orange cap.

**Error Analysis:** The lane-to-cap color mapping is inaccurate. Specifically, Lane 1 is incorrectly described as "yellow cap" when it should be "black cap" and Lane 3 is mistakenly given as "red cap" instead of "white cap". These errors lead to an incorrect overall description of the race outcome.

**Example 4**

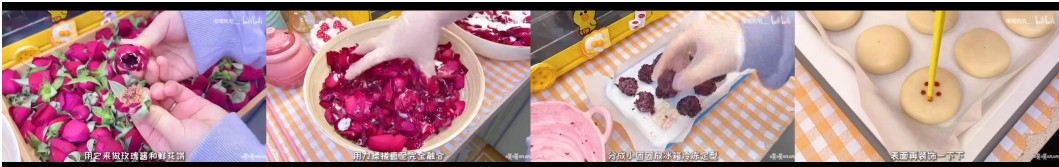

**Instruction:** According to the video, how many kinds of ingredients are in the filling of one type of flower pastry, what are they?

**Generated Caption:** The filling of one type of flower pastry contains three kinds of ingredients: rose petals, sugar, and lemon juice.

**Error Analysis:** The response is incomplete: it fails to list all six ingredients for the flower pastry filling. It provides only a subset and omits several items and descriptive notes. This likely stems from relying on a single segment rather than synthesizing evidence across the entire video. A correct answer should aggregate information from all relevant sections and explicitly enumerate all six ingredients, each accompanied by a brief description.

**Example 5**

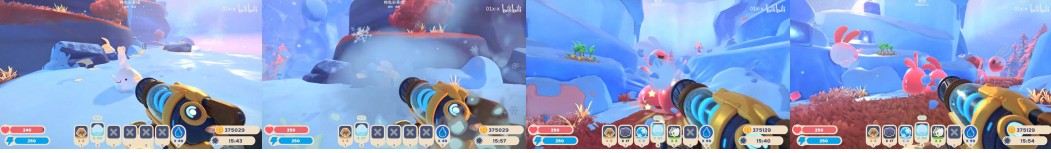

**Instruction:** Describe the video content in chronological order using 3 simple sentences. Focus on the player's actions and the rabbit-like creatures' reactions.

**Generated Caption:** The player uses a water gun to chase and capture fluffy white rabbits in a snowy landscape. The rabbits scatter and hide behind rocks as the player aims and shoots. The scene shifts to a forest where more rabbits appear, some turning pink when hit. The player continues to shoot, capturing several rabbits while the environment changes slightly.

**Error Analysis:** The instructions required a three-sentence response, but the answer included four. In addition, the description was inaccurate—for example, the rabbit is not hiding behind the rock but on top of it in the video.

