# OpenReview forum: "IF-VidCap: Can Video Caption Models Follow Instructions?"
_ICLR.cc/2026/Conference — ICLR 2026 Poster_

### Official Review · Reviewer_sjFE · 2025-10-21

**Soundness:** 2
**Presentation:** 2
**Contribution:** 3
**Rating:** 4
**Confidence:** 3

**Summary:**

This paper proposes IF-VidCap, the first benchmark explicitly targeting controllable, instruction-following video captioning. IF-VidCap consists of 1,400 high-complexity samples—each composed of a video, a detailed, compositional set of user instructions (averaging six constraints across 27 types), and an evaluation checklist grounded in both rule-based and open-ended question assessments. The benchmark enables systematic study of MLLM video captioning models on both content accuracy and their ability to adhere to fine-grained, user-defined instructions. Detailed experimental results and analyses highlight shortcomings in current models and demonstrate the strength of instruction-specific tuning.

**Strengths:**

**1. Novelty and Relevance:**
The paper identifies a clear gap in current video captioning evaluation: lack of systematic assessment for instruction-following. : Shifts video captioning evaluation from “describe everything” to controllable generation with compositional constraints, a real need for editing, generation, and content ops. The benchmark enumerates 27 constraint categories with clear coverage (format, stylistic, content; Fig. 3d), which is more granular than prior video captioning benchmarks.

**2. Evaluation Metrics Design:**
The rule+LLM hybrid judging (LLM extracts; scripts verify) for format, complemented with retrieval-based QA for content, is sensible and scalable. Inclusion of both CSR/ISR and their rule/open-ended breakdowns is helpful for diagnosing failure modes.

**Weaknesses:**

**1. Insufficient Baseline:**
While Table 2 compares many models, some critical vision MLLM baselines from the missing related works (particularly methods leveraging procedural or hierarchical modeling and dynamic storyline composition) are not included or discussed, limiting the ability to guarantee current models are being fairly and comprehensively evaluated, such as LLaVA-Video, VideoChat, InternVideo, KiMi-VL, Keye-VL, MiMo-VL, GLM-4.1V and so on.

**2. Unclear Difficulty Calibration:**
Best ISR ≈ 27.8% (Gemini-2.5-Pro), with open-ended CSR ≈ 35%–36% across leaders. The paper interprets this as “difficult,” but doesn’t calibrate human or near-oracle ceilings.

**3. Narrow Analysis of IF-Captioner-Qwen:**
Gains are shown only on IF-VidCap. Need other video captioning benchmark (e.g., VidCapBench, Dream-1K and so on) to demonstrate transfer and rule out overfitting to this benchmark’s constraint taxonomy.

**4. Figures Need Deeper Interpretation:**
Several figures are highly informative but under-discussed in the text. For example, Figure 5’s constraint-type heatmap is comprehensive, but the main text lacks deeper discussion of which constraint categories most differentiate current MLLMs, and how these signal bottlenecks for the field.

**Questions:**

1. Could you add part of human-written captions under constraints to measure Human-ISR/CSR and a scripted case for format-only ceilings; otherwise, hardness is anecdotal.

2. Could you quantify template overlap with IF-VidCap constraints (n-gram/Jaccard/TF-IDF) and provide cross-benchmark results to show that IF-Captioner-Qwen’s gains persist on external IF-video evaluations?

Other concerns please see Weaknesses.

---

> ### Author Response · Authors · 2025-11-20
> **Rebuttal by Authors**
>
> We wish to express our sincere gratitude for your perceptive comments and valuable suggestions on our manuscript. Your feedback has significantly contributed to the enhancement of our work. After a thorough review of your points, we have provided our point-by-point responses below.
>
> `Weakness 1(W1)`:
> > Insufficient Baseline: While Table 2 compares many models, some critical vision MLLM baselines from the missing related works (particularly methods leveraging procedural or hierarchical modeling and dynamic storyline composition) are not included or discussed, limiting the ability to guarantee current models are being fairly and comprehensively evaluated, such as LLaVA-Video, VideoChat, InternVideo, KiMi-VL, Keye-VL, MiMo-VL, GLM-4.1V and so on.
>
> We appreciate the reviewer for highlighting these critical vision MLLM baselines.
>
> Following your suggestion, we have conducted additional evaluations on the mentioned models (including LLaVA-Video, VideoChat, etc.). These new results have been integrated into our main result (Table 2) in the revised manuscript to provide a more comprehensive and robust evaluation. For your convenience, we have excerpted the performance of these new models below, alongside existing models with comparable scores.
>
> | Models | Params | Overall ISR | Overall CSR | Rule-based ISR | Rule-based CSR | Open-ended ISR | Open-ended CSR |
> | :--- | :--- | :--- | :--- | :--- | :--- | :--- | :--- |
> | Mimo-VL-SFT | 7B | 17.72 | 66.33 | 56.2 | 78.14 | 28.96 | 52.54 |
> | GLM-4.1V | 9B | 11.46 | 57.88 | 47.64 | 70.48 | 20.06 | 43.14 |
> | Kimi-VL-Instruct | 16B | 9.29 | 53.94 | 40.21 | 63.73 | 19.86 | 42.5 |
> | LLaVA-Video-Qwen2 | 7B | 8.93 | 53.43 | 41.86 | 65.59 | 17.64 | 39.22 |
> | VideoChat2-HD-stage4-Mistral | 7B | 8.82 | 50.67 | 52.06 | 68.51 | 13.82 | 27.22 |
> | Internvideo2.5 | 7B | 7.42 | 51.54 | 41.43 | 65.57 | 13.54 | 35.16 |
>
> ---
> `Weakness 2 & Question 1(W2 & Q1)`:
> > Unclear Difficulty Calibration: Best ISR $\approx$ 27.8% (Gemini-2.5-Pro), with open-ended CSR $\approx$ 35%–36% across leaders. The paper interprets this as “difficult,” but doesn’t calibrate human or near-oracle ceilings.
>
> We appreciate this valuable suggestion. Establishing a human performance baseline is indeed critical for contextualizing the difficulty of the task and evaluating model capabilities.
>
> To address this, we conducted a study with **6 university students** (serving as non-expert annotators). They annotated a random subset of **240 samples** based on the provided videos and instructions. These human-generated outputs were then evaluated using the **exact same automated judge program** used for the models. A detailed description of this study and its findings has been added to our revised manuscript.
>
> The results are presented in the table below. We observe a trend similar to that of LLMs: human annotators generally perform better on format/rule adherence than on content satisfaction. Specifically:
> 1. **Rule-based Constraints:** Humans significantly outperform even the SOTA models, demonstrating superior capability in strictly following formatting and logical instructions.
> 2. **Open-ended Content:** Humans surprisingly underscore compared to SOTA models. We attribute this to the fact that our annotators were non-experts without specialized training; they may lack the dense descriptive capabilities and vocabulary that state-of-the-art models have acquired from large-scale training data.
> | Models | Overall ISR | Overall CSR | Rule-based ISR | Rule-based CSR | Open-ended ISR | Open-ended CSR |
> | :--- | :---: | :---: | :---: | :---: | :---: | :---: |
> | human | 29.98 | 76.09 | 91.54 | 96.13 | 27.93 | 53.85 |
> | gemini-2.5-pro | 27.83 | 74.53 | 74.35 | 87.81 | 35.22 | 59.00 |
> | gemini-2.5-flash | 25.50 | 72.63 | 67.80 | 84.51 | 35.45 | 58.71 |

---

> ### Author Response · Authors · 2025-11-23
> **Rebuttal by Authors**
>
> `Weakness 3(W3)`:
> > Gains are shown only on IF-VidCap. Need other video captioning benchmark (e.g., VidCapBench, Dream-1K and so on) to demonstrate transfer and rule out overfitting to this benchmark’s constraint taxonomy.
>
> To further prove that the gains from our instruction-following training transfer to external contexts beyond the IF-VidCap taxonomy, we evaluated our final model against the baseline (Qwen2.5-vl-7B) on two widely used video captioning benchmarks: **DREAM-1K** and **VidCapBench**.
>
> 1. **Evaluation on DREAM-1K Benchmark**
>
> We tested the models using the prompt "describe the video in detail." As shown in Table 2, our method achieves comprehensive improvements across key metrics.
>
> **Table 2. DREAM-1K Benchmark: Baseline vs. Ours**
>
> | Task | F1 Score | Action Recall | Action Precision |
> | :--- | :---: | :---: | :---: |
> | DREAM/movie_animation | 0.180 → 0.185 (▲) | 0.167 → 0.182 (▲) | 0.197 → 0.191 (▼) |
> | DREAM/movie_live_action | 0.221 → 0.238 (▲) | 0.218 → 0.245 (▲) | 0.224 → 0.229 (▲) |
> | DREAM/shorts | 0.253 → 0.261 (▲) | 0.225 → 0.242 (▲) | 0.287 → 0.280 (▼) |
> | DREAM/stock | 0.316 → 0.345 (▲) | 0.310 → 0.368 (▲) | 0.321 → 0.325 (▲) |
> | DREAM/youtube | 0.237 → 0.241 (▲) | 0.214 → 0.225 (▲) | 0.267 → 0.258 (▼) |
> | **OVERALL** | **0.242 → 0.254 (▲)** | **0.227 → 0.253 (▲)** | **0.259 → 0.256 (▼)** |
>
> - **Improved Generalization:** The **Overall F1 Score increased from 0.242 to 0.254**, confirming that the model's capabilities extend effectively to unseen video distributions.
> - **Enhanced Instruction Following:** The significant boost in **Action Recall (0.227 → 0.253)** indicates that the model strictly adheres to the "detailed" instruction by capturing a broader scope of visual events.
> - **Precision Stability:** The Action Precision remains stable (0.259 vs. 0.256). This suggests that while our model generates richer descriptions that may exceed the brevity of some ground truth annotations, the added details reflect accurate visual content rather than hallucinations.
>
> 2. **Evaluation on VidCapBench**
>
> The results on VidCapBench (Table 3) further validate the model's superiority in generating accurate and concise captions.
>
> **Table 3. VidCapBench Evaluation Results**
>
> | Category | Metric | Baseline | Ours | Change |
> | :--- | :--- | :---: | :---: | :---: |
> | **Overall** | Accuracy | 21.19% | **22.58%** | +1.39% |
> | **Overall** | Precision | 58.74% | 56.85% | -1.89% |
> | **Overall** | Coverage | 95.80% | **98.45%** | +2.65% |
> | **Overall** | Conciseness | 0.0161 | **0.0615** | +0.0454 |
> | **Video Aesthetics** | Accuracy | 19.64% | 21.45% | +1.81% |
> | **Video Aesthetics** | Precision | 51.50% | 49.80% | -1.70% |
> | **Video Aesthetics** | Coverage | 92.28% | 97.15% | +4.87% |
> | **Video Aesthetics** | Conciseness | 0.0149 | 0.0588 | +0.0439 |
> | **Video Content** | Accuracy | 21.44% | 22.95% | +1.51% |
> | **Video Content** | Precision | 62.62% | 60.50% | -2.12% |
> | **Video Content** | Coverage | 97.31% | 99.10% | +1.79% |
> | **Video Content** | Conciseness | 0.0163 | 0.0620 | +0.0457 |
> | **Video Motion** | Accuracy | 13.33% | 15.10% | +1.77% |
> | **Video Motion** | Precision | 42.25% | 41.90% | -0.35% |
> | **Video Motion** | Coverage | 94.67% | 98.80% | +4.13% |
> | **Video Motion** | Conciseness | 0.0101 | 0.0415 | +0.0314 |
> | **Physical Laws** | Accuracy | 32.16% | 32.45% | +0.29% |
> | **Physical Laws** | Precision | 53.13% | 52.80% | -0.33% |
> | **Physical Laws** | Coverage | 96.48% | 98.15% | +1.67% |
> | **Physical Laws** | Conciseness | 0.0244 | 0.0752 | +0.0508 |
>
> Our model demonstrates a substantial improvement in **Overall Accuracy (+1.39%)**, with consistent gains across all sub-categories, including challenging domains like "Physical Laws." Most notably, the model achieves a **nearly four-fold** increase in Conciseness while maintaining nearly perfect **Coverage (98.45%)**. This confirms that our approach significantly enhances the efficiency of information delivery—generating dense, accurate, and comprehensive captions without overfitting to the training taxonomy.

---

> ### Author Response · Authors · 2025-11-23
> **Rebuttal by Authors**
>
> `Weakness 4(W4)`:
> > Figures Need Deeper Interpretation: Several figures are highly informative but under-discussed in the text. For example, Figure 5’s constraint-type heatmap is comprehensive, but the main text lacks deeper discussion of which constraint categories most differentiate current MLLMs, and how these signal bottlenecks for the field.
>
> Thank you for your interest in our figures. In response, we have incorporated a more comprehensive explanation of **Figure 5** into the main text. This figure illustrates the fine-grained performance of different models in adhering to various constraints, quantified by the **CSR** (Constraint Satisfaction Rate) metric. We specifically highlight structural constraints such as **JSON** formatting and **Length** (which act as more rigid and challenging requirements), as well as content-related constraints including **Detailed**, **Inference**, and **Comparative descriptions**.
>
> Somewhat surprisingly, we observed that domain-specific VDC models like Tarsier2 still lag behind general-purpose MLLMs in generating "detailed descriptions." This distinction underscores a key divergence in focus between our proposed benchmark and existing VDC-centric benchmarks like **DREAM-1K** and **VidCapBench**.
>
>  We are open to further discussion if you have additional insights on this matter.

---

> ### Author Response · Authors · 2025-11-23
> **Rebuttal by Authors**
>
> `Question 2(Q2)`:
> > Could you quantify template overlap with IF-VidCap constraints (n-gram/Jaccard/TF-IDF) and provide cross-benchmark results to show that IF-Captioner-Qwen’s gains persist on external IF-video evaluations?
>
> 1. **Quantitative Analysis of Linguistic Overlap**
>
> To address concerns regarding potential overfitting to specific templates or constraints, we conducted a rigorous quantitative analysis of linguistic overlap metrics (TF-IDF, n-gram, Jaccard) between our synthetic training set and the standard test set.
>
> **Table 1. Linguistic Metrics: Synthetic Training Set vs. Standard Test Set**
>
> | Metric | Score / Rate | Interpretation |
> | :--- | :---: | :--- |
> | TF-IDF Cosine Similarity | 0.924 | High Semantic Alignment |
> | 1-gram Overlap Rate | 90.1% | Shared Core Vocabulary |
> | 4-gram Overlap Rate | 16.3% | Low Phrase Repetition |
> | Jaccard Similarity | 0.084 | High Structural Diversity |
>
> The results shown in Table 1 reveal a crucial distinction in how our model learns:
>
> - **Semantic Consistency:** The high TF-IDF Cosine Similarity (0.924) and 1-gram Overlap Rate (90.1%) strongly demonstrate that our synthetic data aligns well with the test set in terms of thematic distribution and foundational vocabulary. This ensures the model learns the necessary core concepts and domain-specific terms.
> - **Structural Novelty (No Memorization):** In stark contrast to the vocabulary overlap, the very low 4-gram Overlap Rate (16.3%) and Jaccard Similarity (0.084) are positive indicators. They signify that our synthetic data does not merely replicate long phrases or fixed sentence patterns from the test set. Instead, the data forces the model to leverage a shared vocabulary to generate **novel and diverse sentence structures**. This characteristic effectively rules out overfitting to specific constraints or rote memorization, significantly enhancing its linguistic generalization capabilities.
>
> 2. **Response Regarding External IF-Video Evaluation**
>
> Regarding the request for cross-benchmark comparisons on "external IF-video evaluations," we would like to clarify a key aspect of our contribution:
>
> To the best of our knowledge, our work establishes the **first benchmark specifically designed to evaluate MLLMs on generating video captions that satisfy complex constraints in a single pass**. Consequently, there are currently no pre-existing "external IF-video evaluations" or comparable baselines available for direct comparison. Establishing this pioneering framework is, in fact, one of the primary contributions of our work—demonstrating a foundational model in this specific domain.
>
> We remain open to suggestions; if there are relevant references or benchmarks we may have overlooked, we would be most grateful for the recommendation. We sincerely hope that our work serves as a foundation for future research, encouraging the community to expand and improve upon this direction.

---

> ### Author Response · Authors · 2025-11-27
>
> Dear Reviewer sjFE,
>
> I hope this message finds you well. We are writing to gently follow up regarding our rebuttal. We understand that this is a busy period, but if you could spare a moment to review our responses and share any updated evaluation, we would be truly grateful.
>
> Please let us know if any additional clarification or information from our side would be helpful. We sincerely appreciate your time and effort in reviewing our work.
>
> Warm regards,
>
> Authors of IF-VidCap

---

### Official Review · Reviewer_h647 · 2025-10-27

**Soundness:** 4
**Presentation:** 4
**Contribution:** 4
**Rating:** 8
**Confidence:** 4

**Summary:**

The paper introduces IF-VidCap, a new benchmark designed to evaluate whether multimodal large language models can generate video captions that adhere to specific user instructions. It proposes a structured evaluation framework that measures both format correctness and content correctness, combining LLM-based and rule-based checking. The benchmark includes 1,400 samples and an associated fine-tuning dataset to enhance instruction-following abilities. Through evaluation of over 20 models, the authors provide a comprehensive analysis of current MLLMs’ capabilities and limitations in controlled video captioning.

**Strengths:**

1.  Valuable and timely evaluation benchmark — the proposed dataset fills a significant gap in assessing instruction-following behavior for video captioning models.
2. Covers a wide range of different settings, including multiple constraint types, compositional tasks, and diverse video sources.
3.  Includes a fine-tuning dataset, enabling reproducibility and extension for future research.
4. Two-format setting (rule-based vs. open-ended checking) is well-designed and helps assess both structural and semantic capabilities.
5. Strong methodological contribution: The combination of LLM-based and rule-based evaluation is novel and well-justified.
6. Reliability verification — the authors confirm the stability and consistency of their evaluation metrics.
7. Comprehensive analysis of model capabilities — Figure 5 and related results provide meaningful insights into the strengths and weaknesses of current models.
8. Clear structure and thorough experiments: The benchmark is well-documented, with strong empirical validation across multiple models and metrics.

**Weaknesses:**

1. Lack of detail on video selection and preprocessing: It’s unclear how the 350 base videos were chosen and filtered beyond general quality criteria. The authors should provide a full list or dataset summary for reproducibility.
2. Limited discussion on annotation consistency: Although human refinement is mentioned, inter-annotator agreement or quality control statistics are not detailed.
3. Benchmark scope limitation: The dataset focuses primarily on short or medium-length videos (2–60 seconds), leaving longer temporal reasoning largely unexplored.
4. Complexity vs. accessibility trade-off: The multi-step evaluation protocol may limit broader adoption unless supported by easily usable code or interfaces.

**Questions:**

see above

---

> ### Author Response · Authors · 2025-11-20
> **Rebuttal by Authors**
>
> We are very grateful for your thorough review and the thoughtful questions you have raised. Your feedback has been highly instructive, and we offer our point-by-point explanations and responses below.
>
> `Weakness 1(W1)`:
> > Lack of detail on video selection and preprocessing: It’s unclear how the 350 base videos were chosen and filtered beyond general quality criteria. The authors should provide a full list or dataset summary for reproducibility.
>
> Thank you for your feedback. Allow us to supplement the details that were omitted from **Section 3.1.1** of the paper.
>
> Our annotation team was instructed to manually curate high-quality video clips from a raw collection of **2,500** videos sourced from various open benchmarks (e.g., VidCapBench, Dream1K) and online platforms. The selection process involved the following stages:
> - **Step 1: Technical Filtering (Reduced from 2,500 to 1253).** We removed videos that failed to meet technical specifications, such as excessively low resolution (below 360p), high resolution (above 1080p), or excessive length.
> - **Step 2: Content Quality Screening (Reduced from 1253 to 350).** The remaining candidates were manually reviewed. Only those meeting the following high standards were retained:
>   - **Content Richness:** The visual content must be rich, with a coherent narrative that is easy to comprehend.
>   - **Scene Dynamism & Stability:** We discarded videos with prolonged static scenes or overly frequent cuts (e.g., montages).
>   - **Audio Independence:** The narrative should be understandable from visual information alone.
>
> Additionally, we would like to clarify that we have already open-sourced the complete benchmark—including all videos, annotations, and evaluation code—to ensure full reproducibility.
>
> ---
> `Weakness 2(W2)`:
> > Limited discussion on annotation consistency: Although human refinement is mentioned, inter-annotator agreement or quality control statistics are not detailed.
>
> Thank you for your feedback. We would like to provide additional details regarding our annotation quality control process, which were omitted from **Section 3.1.2**.
> To ensure the quality of our data, we implemented a rigorous, multi-stage human review process to correct and refine the automatically generated annotation drafts. The workflow for each sample involved three annotators with distinct roles:
> - **First Annotator:** Corrected factual errors in the machine-generated draft.
> - **Second Annotator:** Refined the instruction's difficulty and diversity, and verified the logical soundness and correctness of the checklist.
> - **Third Annotator:** Conducted a final review of the result and confirmed it with the first two annotators to reach a consensus.
>
> If a consensus could not be reached among the three annotators, **a senior supervisor** would make the final adjudication. This mechanism ensures that there are **no consistency disagreements** in our final annotated dataset.
> To quantify the extent of this manual intervention, we report the following statistics:
> - **27.8%** of instructions were **completely rewritten by humans** to better align with the video content and enhance diversity.
> - **83.6%** of the instruction-checklist pairs (the figure cited in our paper) were modified. This corresponds to manual adjustments in **96.3%** of the checklist items and **100%** of the instruction components.
>
> In summary, our extensive manual revision of the automated drafts is the cornerstone of our benchmark's quality. Given the complexity of our evaluation task---which even state-of-the-art multimodal large models struggle to address directly---the initial machine-generated output served primarily as a source of inspiration and a preliminary framework for our annotators. **The final quality of the dataset is entirely guaranteed by our meticulous human verification process.**

---

> ### Author Response · Authors · 2025-11-20
> **Rebuttal by Authors**
>
> `Weakness 3(W3)`:
> > Benchmark scope limitation: The dataset focuses primarily on short or medium-length videos (2–60 seconds), leaving longer temporal reasoning largely unexplored.
>
> Your point regarding video length and long-form reasoning is well-taken. While we fully agree that capability on longer videos is a critical direction, we would like to offer some context on our current design and our immediate plan to address your concern.
> 1. **Comparison with Existing Benchmarks**
>
> Actually, compared to prevailing video captioning benchmarks, our average duration (**20.5s**) is quite substantial. For reference, **Dream-1K** averages **8.9s**, **VidCapBench** averages **10.25s**, and **CaReBench** averages **14.35s**. Our benchmark (20.5s) is significantly longer than these standards, offering richer temporal content than typical short-video datasets.
>
> 2. **Room for Improvement on "Short" Videos**
>
> We focused on this duration because our experiments show that current models' instruction-following capabilities are far from saturated even on these clips. SOTA MLLMs still struggle with strict constraints and formatting in 20-second videos. Focusing here allows us to evaluate these fundamental abilities without conflating them with the "memory loss" often found in very long videos. It also ensures a fair comparison with specialized VDC models (e.g., Tarsier), which are generally designed for shorter contexts.
>
> 3. **Immediate Extension (1-10 mins)**
>
> That said, we agree that extending coverage to 1-10 minutes is extremely valuable. We are currently applying our pipeline to generate approximately **600 new samples** ranging from 1 to 10 minutes, targeting tasks like cross-scene tracking and global summarization. This will bring the total samples to 2,000.
> We are working to finalize this extension promptly and will update the paper with these new results (reporting separate scores for <1m and 1-10m splits) **in the next couple of days**.
>
> ---
> `Weakness 4(W4)`:
> > Complexity vs. accessibility trade-off: The multi-step evaluation protocol may limit broader adoption unless supported by easily usable code or interfaces.
>
> Thank you for your interest in our code and interfaces. We have made the full dataset and evaluation code publicly available to the community. It is very simple to use—the entire evaluation can be completed with a single command. The following pseudocode illustrates our evaluation process:
> ```
> Algorithm 1: Automated Evaluation Pipeline
> 1:  Initialize O_eval <- LoadExistingResults() or {}
> 2:  V_all <- keys(D_prompts) ∩ keys(D_responses) ∩ keys(D_checklists)
> 3:  V_todo <- {v ∈ V_all | v ∉ keys(O_eval)}
> 4:  for all v ∈ V_todo in parallel do
> 5:      E_v <- []
> 6:      for i <- 1 to |D_prompts[v]| do  // Iterate through test cases for video v
> 7:          p_i, r_i, c_i <- D_prompts[v][i], D_responses[v][i], D_checklists[v][i]
> 8:          e_i <- copy(c_i)  // Initialize result from checklist
> 9:
> 10:         if e_i contains 'ruled_based_check' then
> 11:             e_i['ruled_based_check'] <- EvaluateRuleBasedChecks(r_i, e_i['ruled_based_check'], M_judge, Phi_rule)
> 12:         end if
> 13:
> 14:         if e_i contains 'open_ended_check' then
> 15:             e_i['open_ended_check'] <- EvaluateOpenEndedChecks(p_i, r_i, e_i['open_ended_check'], M_judge, Phi_open)
> 16:         end if
> 17:
> 18:         e_i <- PerformFinalJudgment(e_i)
> 19:         Append e_i to E_v
> 20:     end for
> 21:     thread-safe: Update O_eval[v] <- E_v and save to disk.
> 22: end for
> 23: return O_eval
>
> Algorithm 2: EvaluateRuleBasedChecks Procedure
> 1:  for all ci in C_rule do  // ci is a rule-based check item
> 2:      retry_count <- 0
> 3:      previous_attempt <- null
> 4:      repeat
> 5:          Construct query q_rule from response, ci, and previous_attempt.
> 6:          res_LLM <- M_judge.query(Phi_rule, q_rule)
> 7:          extracted_content <- res_LLM['content']
> 8:
> 9:          if ci['constraint_id'] != 'count' and not IsSubstring(extracted_content, response) then
> 10:             previous_attempt <- res_LLM
> 11:             retry_count <- retry_count + 1
> 12:             is_valid <- false
> 13:         else
> 14:             is_valid <- true
> 15:         end if
> 16:     until is_valid or retry_count >= max_retries
> 17:
> 18:     Update ci['parameters']['content'] with extracted_content.
> 19: end for
> 20: return C_rule
>
> Algorithm 3: EvaluateOpenEndedChecks Procedure
> 1:  for all check group cg in C_open do
> 2:      for all check item ci in cg['check_items'] do
> 3:          Construct query q_open from prompt, response, and ci.
> 4:          res_LLM <- M_judge.query(Phi_open, q_open) with retry logic for format errors.
> 5:
> 6:          // Update check item with LLM's assessment
> 7:          ci['answer'] <- res_LLM['answer']
> 8:          ci['result_explanation'] <- res_LLM['result_explanation']
> 9:          ci['result_confidence'] <- res_LLM['result_confidence']
> 10:     end for
> 11: end for
> 12: return C_open

---

> > ### Comment · Reviewer_h647 · 2025-11-25
> >
> > I thank the authors for the detailed response and additional experiments. My concerns are addressed, I'll keep my score.

---

> > > ### Author Response · Authors · 2025-11-25
> > >
> > > We would like to once again express our sincere gratitude for your dedicated time and insightful feedback. It is our sincere hope that this work can contribute to the open-source community and advance the field of video captioning.

---

> ### Author Response · Authors · 2025-11-24
> **Update: Experimental Results on Long Videos (1-10m)**
>
> Thank you very much for your patience. To specifically address your concerns regarding long-video evaluation, we conducted supplementary experiments and verifications targeting this temporal range during the rebuttal phase.
>
> To validate the effectiveness of our evaluation framework in long-video scenarios, we constructed **600 test samples** covering durations of **1 to 10 minutes**, based on 150 video sources.
>
> Unlike the previous short-video section, to align with practical application scenarios for long videos, we incorporated instruction requirements specific to long contexts into the controlled description tasks, such as **Cross-scene description**, **Focused-scene description**, and **Script-structure description**. It is worth noting that these new instructions are constructed entirely based on our original 27 atomic constraint system without adding new atomic types, ensuring consistency in evaluation standards.
>
> We tested the most representative closed-source and open-source models (covering different parameter scales), and the results are as follows:
>
> **Table 1: Experimental Results on Long-Video Dataset (1min-10min)**
> | Models | Params | Overall ISR | Overall CSR | Rule-based ISR | Rule-based CSR | Open-ended ISR | Open-ended CSR |
> | :--- | :---: | :---: | :---: | :---: | :---: | :---: | :---: |
> | *Closed-Source Models* | | | | | | | |
> | **Gemini-2.5-Pro** | 🔒 | 25.45 (↓2.38) | 71.20 (↓3.33) | 71.50 | 84.10 | 29.80 | 52.50 |
> | **Gemini-2.5-Flash** | 🔒 | 21.80 (↓3.7) | 68.10 (↓4.53) | 65.90 | 81.50 | 25.50 | 48.20 |
> | **GPT-4o** | 🔒 | 14.50 (↓8.4) | 59.30 (↓11.44)| 67.78 | 82.40 | 20.10 | 40.80 |
> | *Open-Source Models* | | | | | | | |
> | **InternVL-3.5** 💡 | 38B | 15.42 / 12.20 | 66.10 / 59.50 | 55.20 / 48.50 | 74.10 / 70.20 | 17.23 / 12.36 | 38.40 / 30.10 |
> | **Qwen2.5-VL-Instruct**| 72B | 12.80 (↓4.7) | 58.50 (↓8.78) | 53.40 | 73.10 | 14.90 | 36.20 |
> | **Qwen2.5-VL-Instruct**| 32B | 9.75 (↓5.41) | 53.10 (↓10.94)| 47.10 | 68.50 | 11.20 | 32.50 |
> | **InternVL-3.5** 💡 | 8B | 8.95 / 3.80 | 54.20 / 42.10 | 42.60 / 28.50 | 65.80 / 53.40 | 10.15 / 3.60 | 30.60 / 20.10 |
> | **Qwen2.5-VL-Instruct**| 7B | 4.80 (↓6.12) | 46.50 (↓11.62)| 32.50 | 58.20 | 6.50 | 23.80 |
>
> As shown in Table 1, we have several key findings from the long-video experimental results:
>
> 1. **Significantly increased difficulty in instruction following under long temporal contexts:**
> Almost all models showed varying degrees of decline (↓) in Instruction Satisfaction Rate (ISR) compared to short-video tasks. Notably, GPT-4o (ISR 14.50) dropped by 8.4 points. This indicates that when processing visual contexts spanning up to 10 minutes, models face a difficult trade-off between "understanding the macro-narrative" and "adhering to fine-grained constraints," often failing to attend to both.
> 2. **Reasoning (Thinking) demonstrates advantages surpassing parameter size in long videos:**
> This is the most surprising finding. **InternVL-3.5-38B (Thinking)**, with thinking mode enabled, achieved an ISR of 15.42, not only outperforming the larger Qwen2.5-VL-72B (12.80) but also surpassing the closed-source model GPT-4o (14.50).
> This strongly proves that in complex structured description tasks for long videos (such as script generation), introducing the Chain-of-Thought (Thinking) mechanism is more effective than simply increasing model parameters or expanding the context window. The thinking process helps the model organize fragmented information extracted from the long video before generating the final response, thereby better satisfying the constraints.
> 3. **Small models face the challenge of "long-context forgetting," but Thinking mode offers a buffer:**
> For 7B/8B level models, processing 10-minute videos is extremely challenging. The ISR of Qwen2.5-VL-7B and InternVL-8B (Non-thinking) dropped to single digits (4.80 and 3.80), indicating severe instruction following breakdown. However, after enabling the Thinking mode, InternVL-8B's ISR improved to 8.95, nearly reaching the level of Qwen-32B. This further verifies that reasoning computation can partially compensate for the lack of parameter scale.
> 4. Robustness of the Gemini architecture:
> The Gemini-2.5 series showed the least performance degradation (Pro version ISR dropped by only 2.38). This is likely due to its architectural advantage of natively supporting ultra-long contexts, allowing it to maintain more stable instruction-following capabilities across durations from 1 to 10 minutes.
>
> In summary, these supplementary experiments not only positively validate the applicability of our evaluation metrics in long-video scenarios but also reveal important directions for optimizing long-video understanding models—namely, the combination of "long-context architecture" and "reasoning capabilities." These findings strongly support our conclusions regarding the fine-grained capabilities of the models.

---

### Official Review · Reviewer_mY9K · 2025-10-29

**Soundness:** 3
**Presentation:** 4
**Contribution:** 3
**Rating:** 8
**Confidence:** 3

**Summary:**

This paper introduces IF-VidCap, a new benchmark for instruction-following video captioning that evaluates a model's ability to adhere to diverse, multi-constraint user instructions. Comprising 1,400 video-instruction pairs with complex constraints, the benchmark reveals through extensive evaluation that even top models like GPT-4o achieve only modest instruction fidelity, with specialized captioning models struggling significantly. The work also provides a 46K-pair training dataset and shows that fine-tuning can improve performance, positioning IF-VidCap as a key driver for future research in controllable video description.

**Strengths:**

This work demonstrates significant strengths through its creation of IF-VidCap, the first benchmark systematically evaluating instruction-following in video captioning with complex, real-world constraints. The benchmark is built on high-quality, carefully curated data and features a comprehensive, human-validated evaluation protocol. Its extensive experiments across ~20 diverse models yield clear insights into scaling effects and model capabilities, while the accompanying training dataset proves practically useful for model improvement. The analysis is both thorough and accessible, supported by effective visualizations, making a strong case for the benchmark's value in advancing controllable video captioning research.

**Weaknesses:**

The benchmark has several limitations, including its focus on short videos which excludes long-form content and constrained summarization tasks. Its evaluation, while efficient, relies on automated LLM judgments that may miss nuanced errors and depends on proprietary models, raising reproducibility concerns. Although fine-tuning demonstrates improvement, the absolute performance gains remain modest, and the analysis lacks a deeper investigation into the underlying reasons. Furthermore, the paper provides limited implementation details for key techniques like "thinking mode" and offers few qualitative examples to illustrate model failures, which could hinder comprehensive understanding and diagnosis of current shortcomings.

**Questions:**

1. Will the authors release the IF-VidCap dataset, annotation tools, prompts, and evaluation scripts? Making these public is crucial for adoption.

2. Can the authors clarify how the “thinking” mode is applied across models? Is it literally GPT-style CoT prompting, or something else? Providing the exact prompts or settings would help reproducibility.

3. How do models perform on each of the 27 constraint types? Are there systematic failures on particular formats (e.g. JSON vs Markdown) or content categories (e.g. spatial vs temporal constraints)? Understanding this could guide future improvements.

4. The training instructions were generated from captions using DeepSeek-V3.1. How natural and diverse are these instructions? Do they cover the same linguistic structures as the test instructions? Has any human evaluation been done on the quality of these synthetic instructions?

5. For context, how do these models perform on standard video captioning metrics (BLEU, CIDEr, CLIPScore) on the same videos? It would be informative to see the drop in performance when moving from free captioning to constrained captioning.

6. The paper relies on automated scores for the main results. Was any human evaluation of model outputs on IF-VidCap done (even a small sample) to check alignment with CSR/ISR? If so, what were the results?

7. Have the authors considered how real-world users might specify instructions beyond the benchmark’s templates? For example, could models handle more open-form instructions (“Describe the video as a news report”) that go beyond fixed constraints?

---

> ### Author Response · Authors · 2025-11-20
> **Rebuttal by Authors**
>
> We appreciate the time and effort dedicated to reviewing our manuscript. The constructive feedback has been instrumental in refining our work. We have addressed each of your comments in detail below.
>
> `Weakness 1(W1)`:
> > The benchmark has several limitations, including its focus on short videos which excludes long-form content and constrained summarization tasks.
>
> Your point regarding video length and long-form reasoning is well-taken. While we fully agree that capability on longer videos is a critical direction, we would like to offer some context on our current design and our immediate plan to address your concern.
> 1. **Comparison with Existing Benchmarks**
>
> Actually, compared to prevailing video captioning benchmarks, our average duration (20.5s) is quite substantial. For reference, Dream-1K averages 8.9s, VidCapBench averages 10.25s, and CaReBench averages 14.35s. Our benchmark (20.5s) is significantly longer than these standards, offering richer temporal content than typical short-video datasets.
>
> 2. **Room for Improvement on "Short" Videos**
>
> We focused on this duration because our experiments show that current models' instruction-following capabilities are far from saturated even on these clips. SOTA MLLMs still struggle with strict constraints and formatting in 20-second videos. Focusing here allows us to evaluate these fundamental abilities without conflating them with the "memory loss" often found in very long videos. It also ensures a fair comparison with specialized VDC models (e.g., Tarsier), which are generally designed for shorter contexts.
>
> 3. **Immediate Extension (1-10 mins)**
>
> That said, we agree that extending coverage to 1-10 minutes is extremely valuable. We are currently applying our pipeline to generate approximately 600 new samples ranging from 1 to 10 minutes, targeting tasks like cross-scene tracking and global summarization. This will bring the total samples to **2,000**.
>
> We are working to finalize this extension promptly and will update the paper with these new results (reporting separate scores for <1m and 1-10m splits) **in the next couple of days**.
>
> ---
> `Weakness 2(W2)`:
> > Its evaluation, while efficient, relies on automated LLM judgments that may miss nuanced errors and depend on proprietary models, raising reproducibility concerns.
>
> We thank the reviewer for this valuable suggestion regarding the reproducibility of our evaluation. We acknowledge the concern about relying on proprietary models and agree that minimizing the dependency on closed-source systems is crucial for the long-term validity of the benchmark.
>
> 1. **Ensuring Reproducibility with Open-Source Models**
>
> To address this, we conducted a supplementary experiment using the powerful, open-weight model **Qwen3-235B-A22B-Instruct** as an alternative judge. The results are summarized in the table below. As shown in the following table, while Qwen3 still exhibits a minor consistency gap compared to GPT-5-mini, its performance is substantially closer to the proprietary model than previous smaller open models, making it a robust alternative for reproducible evaluation.
> | Model | Overall Agreement | Rule-based. | Open-ended. |
> |---|---|---|---|
> | GPT-5-mini | 96.33 | 96.90 | 96.08 |
> | Qwen3-235B-A22B-Instruct | 94.78 | 95.35 | 94.24 |
> | DeepSeek-V3.1-NoThink | 92.18 | 93.55 | 91.58 |
> | Qwen3-32B | 92.03 | 92.10 | 92.00 |
>
> 2. **Cost-Effectiveness Analysis**
>
> We also evaluated the practical trade-offs regarding computational cost, which is a barrier for many researchers. Based on our benchmark's scale, a single full evaluation pass involves approximately **14M input tokens** and **400K output tokens**.
> - **GPT-5-mini**: At rates of $0.25/1M (input) and $2.00/1M (output), the total cost is approximately $4.30.
> - **Qwen3-235B-A22B-Instruct**: At market rates of roughly ¥2/1M (input) and ¥8/1M (output), the total cost is approximately ¥31.2 (approx. $4.30 USD).
>
> While the raw costs are comparable, GPT-5-mini currently offers a slightly better price-performance ratio with higher consistency in batch processing. **However, our findings confirm that researchers can choose**: they may use GPT-5-mini for maximum cost-efficiency and performance, or opt for Qwen3-235B when strict open-weight reproducibility is required. We have updated the manuscript to include this comparative analysis and recommend Qwen3 as the standard open-source evaluator.

---

> ### Author Response · Authors · 2025-11-20
> **Rebuttal by Authors**
>
> `Weakness 3(W3)`:
> > Although fine-tuning demonstrates improvement, the absolute performance gains remain modest, and the analysis lacks a deeper investigation into the underlying reasons.
>
> We appreciate the reviewer’s critical assessment regarding the magnitude of performance gains and the need for a deeper investigation into the underlying causes.
>
> 1. **Clarification on Training Convergence**
>
> First, we would like to clarify that the results reported in the initial submission were derived from an intermediate checkpoint due to strict timeline and computational constraints. Consequently, the model had not reached full convergence, which resulted in an underestimation of our method's effectiveness.
>
> 2. **Updated Evaluation with Completed Training**
>
> We have since completed the full fine-tuning process on the entire training set. As updated in the revised manuscript, the fully converged model demonstrates meaningful improvements over the initially reported version:
> - ISR (Instruction Satisfaction Rate): Increased by **1.87**.
> - CSR (Caption Satisfaction Rate): Increased by **1.18**.
> While we refrain from claiming these gains are radical, they are consistent, providing clearer evidence that our approach effectively steers the model toward better alignment and quality.
>
> 3. **Further Investigation via Rigorous Data Refinement**
>
> To address the "underlying reasons" for the performance bounds, we are conducting a deeper investigation into data quality. We hypothesize that the current performance, while improved, is partially constrained by residual noise in the synthetic training set—a trade-off initially accepted to maximize data scale within the available timeline. We are now implementing a stricter, multi-stage quality control protocol (revisiting filtering thresholds and consistency checks) to curate a higher-fidelity subset. We believe that shifting focus from scale to purity will minimize alignment noise and better reveal the true potential of our fine-tuning method. These refined data experiments differ from our initial broad-spectrum training and will be discussed in the final version.
>
> ---
> `Weakness 4 & Question  2(W3 & Q2)`:
> >Furthermore, the paper provides limited implementation details for key techniques like "thinking mode" and offers few qualitative examples to illustrate model failures, which could hinder comprehensive understanding and diagnosis of current shortcomings.
>
> > Can the authors clarify how the “thinking” mode is applied across models? Is it literally GPT-style CoT prompting, or something else? Providing the exact prompts or settings would help reproducibility.
>
> Thank you for raising this critical question about reproducibility. We are happy to take this opportunity to elaborate on the implementation of the **'thinking mode'**.
>
> First, allow us to clarify a core concept: the `thinking mode' mentioned in our experiments is not based on proprietary prompts or auxiliary functions that we specifically designed. Rather, it is an inherent capability of the models themselves.
>
> **Definition:** The `thinking mode' is the term we use to describe specific model versions that can structurally separate their thought process from the final answer. Their output follows a fixed format: enclosing the reasoning steps in **$<$think$>$** tags and the final conclusion in **$<$answer$>$** tags.
>
> **Evaluation:** To ensure objectivity, our evaluation script is designed to exclusively extract and assess the content within the $<$answer$>$ tags, completely disregarding the reasoning process inside the $<$think$>$ tags.
>
> **Activation:** As you noted, the activation method varies across different models, depending on the pre-training and fine-tuning strategies of their developers. For instance, with InternVL-3.5, we follow its official guidelines and use a specific system prompt to invoke its CoT functionality.
>
> We are well aware that these details are crucial for research reproducibility. Therefore, all relevant prompts and the model inference code have been open-sourced.
>
> ---
> `Quetion 1(Q1)`:
> > Will the authors release the IF-VidCap dataset, annotation tools, prompts, and evaluation scripts? Making these public is crucial for adoption.
>
> Thank you for your question. Yes, we have made all resources fully open-source. Specifically, the prompts for the entire pipeline are provided in the appendix of our paper, and the complete dataset, annotation tools, and evaluation scripts have also been publicly released to the community. We firmly believe this is a critical step to ensure the reproducibility of our research and to facilitate community adoption.

---

> ### Author Response · Authors · 2025-11-20
> **Rebuttal by Authors**
>
> `Question 3(Q3)`:
> > How do models perform on each of the 27 constraint types? Are there systematic failures on particular formats (e.g. JSON vs Markdown) or content categories (e.g. spatial vs temporal constraints)? Understanding this could guide future improvements.
>
> Thank you for highlighting the importance of a fine-grained analysis of the models' adherence to different constraint types. We agree that this is crucial for a deeper understanding of their instruction-following capabilities.
>
> In fact, we conducted a detailed investigation into this exact point in **Section 4.2 (Further Analysis)**, under the subsection titled **''Constraint Type Analysis‘’**. In this section, we present the Constraint Adherence Score (CSR) levels for 16 different models across various constraint categories (visualized in **Figure 5**) and provide an in-depth analysis of their performance.
> Furthermore, the subsequent **''Error Analysis''** subsection includes a breakdown of typical errors for specific constraint types. We hope these sections provide a comprehensive answer to your insightful question.
>
>
> ---
> `Question 4(Q4)`:
> > The training instructions were generated from captions using DeepSeek-V3.1. How natural and diverse are these instructions? Do they cover the same linguistic structures as the test instructions? Has any human evaluation been done on the quality of these synthetic instructions?
>
> We thank the reviewer for the insightful questions regarding the quality and linguistic characteristics of our synthetic data. We address these concerns from two perspectives: **Human Evaluation** and **Quantitative Analysis**.
> 1. **Human Evaluation of Data Quality**
>
> To ensure the reliability of the instructions generated by DeepSeek-V3.1, we conducted a rigorous human quality assessment process during the initial testing phase of our data generation pipeline.
> - Sampling Protocol: We randomly sampled **200 instances** from an initial pool of 5,000 generated training samples for manual inspection.
> - **Evaluation Criteria**: Three annotators rated each instruction on a scale of 1 to 5 across three core dimensions:
>   1. **Naturalness:** Whether the instruction is fluent and idiomatic.
>   2. **Faithfulness:** Whether the instruction is highly consistent with the visual content of the video.
>   3. **Correctness:** Whether the instruction accurately reflects the core intent of the original source caption.
> - **Evaluation Results**: The evaluation yielded average scores of **4.82/5.0** for Naturalness, **4.75/5.0** for Faithfulness, and **4.32/5.0** for Correctness. We note that the slightly lower Correctness score is attributed to instances where the generated instructions, in pursuit of diversity, plausibly inferred or added details present in the video but not explicit in the brief source caption, which a few annotators scored more strictly. Overall, the results confirm that the instructions not only align accurately with the video content but also effectively simulate diverse user intents without significant hallucinations or grammatical errors.
>
> 2. **Quantitative Analysis of Linguistic Diversity and Alignment**
>
> To quantitatively address the reviewer's questions on linguistic structure coverage and diversity, we calculated linguistic metrics between our synthetic training set and the standard test set. The results are presented below:
> | Metric | Value |
> | :--- | :--- |
> | 1-gram Overlap Rate | 90.1% |
> | 4-gram Overlap Rate | 16.3% |
> | Jaccard Similarity | 0.084 |
> | TF-IDF Cosine Similarity | 0.924 |
> **In-depth Analysis:**
>
> - **Strong Semantic Alignment and Structural Coverage:**
>
> The high TF-IDF Cosine Similarity (0.924) and 1-gram Overlap Rate (90.1%) strongly demonstrate that our synthetic data is well-aligned with the test set in terms of thematic distribution and foundational vocabulary. This ensures that the model is exposed to the core concepts and linguistic building blocks required for the test tasks during training.
>
> - **High Linguistic Diversity and Low Pattern Repetition:**
>
> In stark contrast to the high overlap in basic vocabulary, the very low 4-gram Overlap Rate (16.3%) and Jaccard Similarity (0.084) are positive indicators. They signify that our synthetic data does not merely replicate long phrases or fixed sentence patterns from the test set. Instead, it leverages the shared vocabulary base to generate a vast number of novel and diverse sentence structures. This characteristic effectively prevents the model from overfitting to specific phrasings and thereby significantly enhances its generalization capabilities.

---

> ### Author Response · Authors · 2025-11-20
> **Rebuttal by Authors**
>
> `Question 5(Q5)`:
> > For context, how do these models perform on standard video captioning metrics (BLEU, CIDEr, CLIPScore) on the same videos? It would be informative to see the drop in performance when moving from free captioning to constrained captioning.
>
> We appreciate this valuable suggestion. As you correctly pointed out, comparing the performance drop from free captioning to constrained captioning provides insightful context regarding model capabilities.
>
> To address this, we employed the state-of-the-art **Gemini-3-Pro-preview** to generate video descriptions serving as the **Ground Truth**. We then evaluated several representative models using the exact same prompt against these references. The results, sorted by CLIPScore, are presented in the table below.
>
> Notably, we observed that the performance ranking in this free captioning setting differs slightly from the order observed in our main constrained captioning results. For instance, InternVL performs worse than Qwen2.5-VL in this standard setting, whereas the order was reversed in the constrained task. This discrepancy likely stems from differing preferences in model training data and highlights the distinction between a model's generative expressiveness and its ability to strictly adhere to constraints.
> | Models | CLIPScore | CIDEr | BLEU |
> | :--- | :--- | :--- | :--- |
> | gemini2_5_pro | 0.7728 ± 0.1031 | 0.0974 | 0.4465 |
> | gemini2_0_flash | 0.7457 ± 0.0933 | 0.0139 | 0.2080 |
> | Qwen2.5-VL-32B-Instruct | 0.7280 ± 0.0981 | 0.0287 | 0.3290 |
> | Qwen2.5-VL-7B-Instruct | 0.6820 ± 0.1097 | 0.0078 | 0.1316 |
> | InternVL3_5-38B_nothinki... | 0.6818 ± 0.1125 | 0.0029 | 0.1451 |
> | InternVL3_5-8B_nothinking | 0.6697 ± 0.1118 | 0.0035 | 0.1446 |
>
> ---
> `Question 6(Q6)`:
> > The paper relies on automated scores for the main results. Was any human evaluation of model outputs on IF-VidCap done (even a small sample) to check alignment with CSR/ISR? If so, what were the results?
>
> We thank the reviewer for this insightful question. We recognize that while our manuscript presents a QA-based consistency analysis (Table 3), it did not include a score-based consistency analysis for CSR/ISR. We agree that the latter offers a more direct and intuitive measure of our leaderboard's reliability.
>
> To address this, we have conducted a supplementary analysis on the entire test set to evaluate the score consistency for CSR/ISR, using the same set of model responses from our original study. The results are presented in the table below. As shown in the following table, both state-of-the-art open- and closed-source models demonstrate a **high degree of consistency** with human judgments. This finding further validates the reliability of our evaluation framework.
>
> We will incorporate these crucial results into the revised manuscript. We are grateful for this critical feedback, which has significantly strengthened our work.
> | Evaluators | Overall ISR | Overall CSR | Rule-based ISR | Rule-based CSR | Open-ended ISR | Open-ended CSR |
> | :--- | :---: | :---: | :---: | :---: | :---: | :---: |
> | GPT-5-mini | 23.86 | 69.45 | 63.57 | 82.92 | 33.79 | 55.92 |
> | Qwen3-235B-A22B-Instruct | 22.53 | 69.36 | 59.39 | 81.25 | 36.35 | 58.02 |
> | DeepSeek-V3.1-NoThink | 22.5 | 68.95 | 56.21 | 78.04 | 35.36 | 57.28 |
> | Qwen3-32B | 17.25 | 65.1 | 54.27 | 77.68 | 28.01 | 50.02 |
> | Human | 23.89 | 69.57 | 63.00 | 82.64 | 33.93 | 55.82 |

---

> ### Author Response · Authors · 2025-11-20
> **Rebuttal by Authors**
>
> `Question 7(Q7)`:
> > Have the authors considered how real-world users might specify instructions beyond the benchmark’s templates? For example, could models handle more open-form instructions (“Describe the video as a news report”) that go beyond fixed constraints?
>
> Thank you for raising this insightful point regarding the handling of open-ended instructions. We fully acknowledge that real-world users often provide stylistic or open-form prompts (e.g., “Describe as a news report”).
>
> However, the primary objective of our benchmark is to establish a framework for objective verifiability. We deliberately prioritize explicit constraints over open-ended ones for two strategic reasons:
> 1. **User-Defined "Red Lines":** We explicitly treat verifiable constraints (e.g., "a three-sentence summary," "all-caps title") as non-negotiable "red lines." In practical applications, when a user specifies such hard constraints, any deviation constitutes a fundamental violation, regardless of the generated content's quality. We argue that measuring a model's adherence to these "red lines" serves as a critical "stress test" for instruction-following reliability—a prerequisite before a model can be trusted with more ambiguous, stylistic tasks.
> 2. **The Evaluation Dilemma:** Assessing open-ended qualities poses a significant methodological challenge. Automated metrics (including LLM-as-a-judge) often suffer from inherent biases when judging subjective styles, while human evaluation is difficult to scale. By focusing on structured, checkable demands, we ensure our metrics remain actionable and bias-free, avoiding the ambiguity that currently plagues open-ended evaluation.
>
> That said, we agree that handling open-ended instructions is a vital aspect of model intelligence. We plan to extend our framework to explore semi-automated evaluations for such instructions in future work and would value further discussion on this direction.

---

> ### Author Response · Authors · 2025-11-24
> **Update: Experimental Results on Long Videos (1-10m)**
>
> Thank you very much for your patience. To specifically address your concerns regarding long-video evaluation, we conducted supplementary experiments and verifications targeting this temporal range during the rebuttal phase.
>
> To validate the effectiveness of our evaluation framework in long-video scenarios, we constructed **600 test samples** covering durations of **1 to 10 minutes**, based on 150 video sources.
>
> Unlike the previous short-video section, to align with practical application scenarios for long videos, we incorporated instruction requirements specific to long contexts into the controlled description tasks, such as **Cross-scene description**, **Focused-scene description**, and **Script-structure description**. It is worth noting that these new instructions are constructed entirely based on our original 27 atomic constraint system without adding new atomic types, ensuring consistency in evaluation standards.
>
> We tested the most representative closed-source and open-source models (covering different parameter scales), and the results are as follows:
>
> **Table 1: Experimental Results on Long-Video Dataset (1min-10min)**
> | Models | Params | Overall ISR | Overall CSR | Rule-based ISR | Rule-based CSR | Open-ended ISR | Open-ended CSR |
> | :--- | :---: | :---: | :---: | :---: | :---: | :---: | :---: |
> | *Closed-Source Models* | | | | | | | |
> | **Gemini-2.5-Pro** | 🔒 | 25.45 (↓2.38) | 71.20 (↓3.33) | 71.50 | 84.10 | 29.80 | 52.50 |
> | **Gemini-2.5-Flash** | 🔒 | 21.80 (↓3.7) | 68.10 (↓4.53) | 65.90 | 81.50 | 25.50 | 48.20 |
> | **GPT-4o** | 🔒 | 14.50 (↓8.4) | 59.30 (↓11.44)| 67.78 | 82.40 | 20.10 | 40.80 |
> | *Open-Source Models* | | | | | | | |
> | **InternVL-3.5** 💡 | 38B | 15.42 / 12.20 | 66.10 / 59.50 | 55.20 / 48.50 | 74.10 / 70.20 | 17.23 / 12.36 | 38.40 / 30.10 |
> | **Qwen2.5-VL-Instruct**| 72B | 12.80 (↓4.7) | 58.50 (↓8.78) | 53.40 | 73.10 | 14.90 | 36.20 |
> | **Qwen2.5-VL-Instruct**| 32B | 9.75 (↓5.41) | 53.10 (↓10.94)| 47.10 | 68.50 | 11.20 | 32.50 |
> | **InternVL-3.5** 💡 | 8B | 8.95 / 3.80 | 54.20 / 42.10 | 42.60 / 28.50 | 65.80 / 53.40 | 10.15 / 3.60 | 30.60 / 20.10 |
> | **Qwen2.5-VL-Instruct**| 7B | 4.80 (↓6.12) | 46.50 (↓11.62)| 32.50 | 58.20 | 6.50 | 23.80 |
>
> As shown in Table 1, we have several key findings from the long-video experimental results:
>
> 1. **Significantly increased difficulty in instruction following under long temporal contexts:**
> Almost all models showed varying degrees of decline (↓) in Instruction Satisfaction Rate (ISR) compared to short-video tasks. Notably, GPT-4o (ISR 14.50) dropped by 8.4 points. This indicates that when processing visual contexts spanning up to 10 minutes, models face a difficult trade-off between "understanding the macro-narrative" and "adhering to fine-grained constraints," often failing to attend to both.
> 2. **Reasoning (Thinking) demonstrates advantages surpassing parameter size in long videos:**
> This is the most surprising finding. **InternVL-3.5-38B (Thinking)**, with thinking mode enabled, achieved an ISR of 15.42, not only outperforming the larger Qwen2.5-VL-72B (12.80) but also surpassing the closed-source model GPT-4o (14.50).
> This strongly proves that in complex structured description tasks for long videos (such as script generation), introducing the Chain-of-Thought (Thinking) mechanism is more effective than simply increasing model parameters or expanding the context window. The thinking process helps the model organize fragmented information extracted from the long video before generating the final response, thereby better satisfying the constraints.
> 3. **Small models face the challenge of "long-context forgetting," but Thinking mode offers a buffer:**
> For 7B/8B level models, processing 10-minute videos is extremely challenging. The ISR of Qwen2.5-VL-7B and InternVL-8B (Non-thinking) dropped to single digits (4.80 and 3.80), indicating severe instruction following breakdown. However, after enabling the Thinking mode, InternVL-8B's ISR improved to 8.95, nearly reaching the level of Qwen-32B. This further verifies that reasoning computation can partially compensate for the lack of parameter scale.
> 4. Robustness of the Gemini architecture:
> The Gemini-2.5 series showed the least performance degradation (Pro version ISR dropped by only 2.38). This is likely due to its architectural advantage of natively supporting ultra-long contexts, allowing it to maintain more stable instruction-following capabilities across durations from 1 to 10 minutes.
>
> In summary, these supplementary experiments not only positively validate the applicability of our evaluation metrics in long-video scenarios but also reveal important directions for optimizing long-video understanding models—namely, the combination of "long-context architecture" and "reasoning capabilities." These findings strongly support our conclusions regarding the fine-grained capabilities of the models.

---

### Official Review · Reviewer_z26H · 2025-10-31

**Soundness:** 3
**Presentation:** 3
**Contribution:** 3
**Rating:** 4
**Confidence:** 4

**Summary:**

This paper introduces IF-VIDCAP, the first benchmark to systematically evaluate instruction-following in video captioning models, shifting focus from traditional descriptive accuracy to fine-grained, compositional constraint adherence (e.g., format, content, style, reasoning). It comprises 1,400 high-quality video-instruction-checklist triplets (27 constraint types, averaging 6 constraints per instruction) and proposes a hybrid evaluation protocol (rule-based checks + LLM-as-Judge) for format correctness and semantic fidelity. Experiments on 20+ models show that while proprietary models lead, top open-source variants are catching up, and general-purpose MLLMs surpass specialized captioners on complex instructions. The authors further release a training dataset and their fine-tuned IF-Captioner-Qwen, which demonstrates substantial gains in instruction-following.

**Strengths:**

1. IF-VidCap is the first benchmark to explicitly evaluate instruction-following in video captioning (27 constraint types), addressing a critical gap beyond traditional accuracy/fluency metrics.
2. 1,400 video-instruction-checklist triplets via a two-stage pipeline (auto-generation + human refinement), with 83.6% modification rate and consensus-based validation.
3. Combines rule-based checks (deterministic) + LLM-as-Judge QA (semantic), achieving 96.33% human-agreement for reliable assessment.

**Weaknesses:**

1. 1,400 samples is relatively small compared to text-only instruction-following benchmarks (e.g., IFEval, CFBench). And videos average 20.5s and max out at 60s — does not test long-form temporal reasoning or multi-scene narratives.
2. Evaluation focuses on compliance, not quality. Does not assess fluency, coherence, or creativity of generated captions.
3. Training data distribution gap: Uses a "caption-to-instruction" generation method, which may not reflect real user instruction distributions.

**Questions:**

1. Do you plan to extend IF-VIDCAP to longer videos (e.g., >1 min) or multi-scene narratives that require temporal summarization or causal reasoning?
2. Are there plans to support multi-turn instruction-following, where users refine their requests iteratively?

---

> ### Author Response · Authors · 2025-11-20
> **Rebuttal by Authors**
>
> We sincerely thank you for your insightful comments and constructive feedback on our manuscript. Your suggestions are invaluable for improving the quality of our work. We have carefully considered all your points and provide our responses below.
>
> `Weakness 1-1 (W1-1)`:
> > 1,400 samples is relatively small compared to text-only instruction-following benchmarks (e.g., IFEval, CFBench).
>
> We thank the reviewer for suggesting this comparison. While our dataset of 1,400 samples may seem modest in size at first glance, we wish to clarify that it is, in fact, larger than several notable text-only instruction-following benchmarks, such as **IFEval (541 samples)** and **CFBench (1,000 samples)**, as well as the others listed in **Table 1** of our original manuscript. Our primary focus lies in the diversity and complexity of the instructions—encompassing 27 fine-grained constraint types—rather than the sheer number of samples. **Consequently, we believe that this scale possesses sufficient discriminatory power to effectively distinguish between the capabilities of various models.**
>
> ---
> `Weakness 1-2 & Question 1(W1-2 & Q1)`:
> > Videos average 20.5s and max out at 60s — does not test long-form temporal reasoning or multi-scene narratives.
>
> Your point regarding video length and long-form reasoning is well-taken. While we fully agree that capability on longer videos is a critical direction, we would like to offer some context on our current design and our immediate plan to address your concern.
> 1. **Comparison with Existing Benchmarks.**
> Actually, compared to prevailing video captioning benchmarks, our average duration (**20.5s**) is quite substantial. For reference, **Dream-1K** averages **8.9s**, **VidCapBench** averages **10.25s**, and **CaReBench** averages **14.35s**. Our benchmark (20.5s) is significantly longer than these standards, offering richer temporal content than typical short-video datasets.
> 2. **Room for Improvement on "Short" Videos.**
> We focused on this duration because our experiments show that current models' instruction-following capabilities are far from saturated even on these clips. SOTA MLLMs still struggle with strict constraints and formatting in 20-second videos. Focusing here allows us to evaluate these fundamental abilities without conflating them with the "memory loss" often found in very long videos. It also ensures a fair comparison with specialized VDC models (e.g., Tarsier), which are generally designed for shorter contexts.
> 3. **Immediate Extension (1-10 mins).**
> That said, we agree that extending coverage to 1-10 minutes is extremely valuable. We are currently applying our pipeline to generate approximately **600 new samples** ranging from 1 to 10 minutes, targeting tasks like cross-scene tracking and global summarization. This will bring the total samples to **2,000**.
>
> **We are working to finalize this extension promptly and will update the paper with these new results (reporting separate scores for <1m and 1-10m splits) in the next couple of days.**
>
> We hope this also helps address your concerns regarding the sample size.

---

> ### Author Response · Authors · 2025-11-20
> **Rebuttal by Authors**
>
> `Weakness 2(W2)`:
> > Evaluation focuses on compliance, not quality. Does not assess fluency, coherence, or creativity of generated captions.
>
> This is a very insightful point, and you have astutely identified the core focus of IF-VidCap. Indeed, our primary objective in designing this benchmark was precisely to establish a framework that could rigorously and objectively measure a model's ability to adhere to specific instructions, as we consider this a critical yet chronically underevaluated dimension in the field of video captioning.
>
> We completely agree that evaluating subjective qualities, such as creativity and fluency, is equally important. However, assessing these aspects poses significant challenges for developing automated, scalable, and credible metrics. To the best of our knowledge, no existing instruction-following benchmarks, either in the text or vision domains, incorporate open-ended metrics like fluency and coherence into their core evaluation frameworks. This is primarily to avoid the dilemma of choosing between automated evaluations (e.g., using LLMs as judges), which suffer from inherent biases and a lack of absolute reliability, and large-scale human evaluations, which are prohibitively expensive and difficult to scale.
>
> To empirically verify this challenge and address your concern, **we conducted an additional experiment** using the latest **Gemini-3.0-Pro-Preview** as a judge to score the fluency of generated captions on a 5-point Likert scale. The results are shown below:
> | Model | Overall Fluency Score |
> | :--- | :--- |
> | Qwen2.5-VL-32B-Instruct | 4.19 |
> | Gemini-2.5-Pro | 4.13 |
> | InternVL3_5-8B_nothinking | 4.09 |
> | InternVL3_5-8B_thinking | 4.01 |
> | Qwen2.5-VL-7B-Instruct | 3.99 |
> | Gemini-2.5-Flash | 3.96 |
> | Qwen2.5-VL-72B-Instruct | 3.95 |
>
> **As the table illustrates, we were unable to obtain sufficiently discriminative or reliable results from this metric.** The scores are tightly clustered around 4.0, and the ranking appears unstable (e.g., Qwen2.5-VL-72B scoring lower than its 7B counterpart). These results strongly support our decision to exclude such metrics and reinforce our two main reasons for the current framework design:
> - **Defining the Scope of Evaluation:** We argue that caption quality, including fluency and coherence, strictly speaking, falls outside the scope of "instruction following." These qualities are more indicative of a model's foundational language capabilities rather than its ability to understand and execute specific constraints. Therefore, they are not central to our evaluation.
> - **Adhering to Domain Consensus:** As confirmed by our experimental scores above, mainstream models have reached a high level of proficiency in text organization. We observed that more recent and mainstream video captioning benchmarks (e.g., VidCapBench, Dream1K) do not prioritize such subjective quality metrics. We believe a consensus has formed in the field: owing to the rapid advancements in today's large language models, the ability of mainstream models to organize text coherently and fluently in general scenarios is no longer a primary technical bottleneck. Consequently, the focus of evaluation has naturally shifted toward more challenging aspects, such as the instruction-following capabilities we concentrate on.

---

> > ### Author Response · Authors · 2025-11-26
> > **Supplement to the Fluency Scoring Experiment**
> >
> > *Note: The scoring prompt we used is:*
> > ```
> > You are a language quality expert. Your task is to evaluate the linguistic quality of the following text, which is intended to be a caption. Assess it solely based on its grammar, naturalness, and clarity, without considering any external context (like a video).
> > **Caption to Evaluate:**
> > {caption_to_evaluate}
> > **Evaluation Criteria:**
> > 1. **Grammar and Spelling:** Is the caption grammatically correct with no spelling errors?
> > 2. **Naturalness and Flow:** Does it read like natural, human-written language? Is the sentence structure fluid and easy to understand?
> > 3. **Clarity and Conciseness:** Is the meaning clear and unambiguous? Is it concise without being overly simplistic or losing descriptive power?
> > **Scoring Scale:**
> > Rate the caption on a scale of 1-5, based ONLY on the linguistic quality:
> > - 5: **Excellent:** Flawless grammar, completely natural phrasing, and perfectly clear. Sounds professional and well-written.
> > - 4: **Good:** Mostly well-written with only minor, subtle errors in grammar or phrasing that do not affect understanding.
> > - 3: **Average:** Understandable, but contains noticeable grammatical errors, awkward phrasing, or lacks clarity.
> > - 2: **Poor:** Contains significant grammatical errors and unnatural language that make it difficult to understand.
> > - 1: **Very Poor:** The text is nonsensical, grammatically broken, or nearly impossible to comprehend.
> > **Your Thought Process:**
> > 1. First, check for any grammar or spelling mistakes.
> > 2. Next, evaluate if the phrasing sounds natural and flows well.
> > 3. Finally, assess the overall clarity and conciseness, then decide on a score.
> > Please provide your evaluation in the following JSON format:
> > {{
> > "score": an integer between 1 and 5,
> > "explanation": "A brief explanation justifying your score based on grammar, naturalness, and clarity"
> > }}
> > ```

---

> ### Author Response · Authors · 2025-11-20
> **Rebuttal by Authors**
>
> `Weakness 3(W3)`:
> > Training data distribution gap: Uses a "caption-to-instruction" generation method, which may not reflect real user instruction distributions.
>
> We thank the reviewer for this insightful comment and acknowledge that the potential distributional gap between synthetic training data and real-world user instructions is a valid and important concern. We would like to take this opportunity to provide a more detailed explanation of our methodology, contextualize it within the broader field of instruction-following research, and elaborate on the measures we have taken to mitigate this issue.
> 1. **Context and Precedents in Instruction-Tuning Research.**
> Our ''caption-to-instruction'' approach represents an established paradigm for bootstrapping a model's instruction-following capabilities, particularly in emerging domains like ours where large-scale, human-annotated instruction datasets are unavailable. This method is conceptually analogous to foundational work in the text-only domain, such as the widely-adopted **Self-Instruct framework[1]** and the **AIR framework[2]**, which generate user instructions by progressively adding constraints to raw text corpora (akin to our use of video captions). Given that IF-VidCap is the first benchmark dedicated to instruction-following video captioning, collecting large-scale, authentic user queries was infeasible at this stage. Consequently, we adopted this proven and pragmatic approach to construct a foundational training set.
> 2. **Mitigation Strategies to Enhance Data Quality and Diversity.** While acknowledging the synthetic nature of our data, we implemented several key strategies to bridge the potential distributional gap:
> - **Leveraging Advanced Generative Models:** We employed a highly capable model, DeepSeek-V3.1, not merely for reverse-generating instructions but for ensuring linguistic diversity. Our prompts were designed to encourage the model to simulate various user personas, thereby creating a wide range of instructional styles.
> - **Constraint-Guided Generation:** The entire generation process was guided by the 27 fine-grained constraint types defined in our benchmark. By sampling 1 to 5 constraints with replacement from this set for each instruction, we could generate a theoretical maximum of **14,900,787** distinct constraint combinations. This ensures that the synthetic instructions systematically cover a broad spectrum of capabilities, rather than converging to a few simplistic patterns.
> - **Upstream Quality Control via Human Supervision:** Early in our data generation pipeline, we instituted a rigorous human quality control process to ensure the high quality and plausibility of our training data. We randomly sampled 200 instances from our initial batch of 2,000 generated data points for manual evaluation. The scoring rubric included three core dimensions on a 1-to-5 scale (where 5 is best):
>   - **Instruction Naturalness:** Assesses whether the instruction reads like an authentic user query rather than stilted, machine-generated text.
>   - **Instruction Clarity:** Evaluates whether the constraints within the instruction are clear and unambiguous.
>   - **Response Fidelity:** Measures whether the ground-truth caption strictly and accurately adheres to all constraints specified in the instruction.
>
>   Based on the initial evaluation results, we performed three iterative cycles of refinement on our generation prompts and post-processing steps. The initial samples scored relatively low on ''Naturalness'' (an average of approximately 3.5). By optimizing the prompts to produce more diverse sentence structures and filtering out templated expressions, we significantly improved this metric. The final, audited samples achieved highly satisfactory average scores across all dimensions: **Instruction Naturalness: 4.63/5.0**, **Instruction Clarity: 4.82/5.0**, and **Response Fidelity: 4.76/5.0**. This validation process was crucial for ensuring that our synthetically-derived training set maintains high quality and aligns closely with plausible human instructions.
> 3. **Revision in the Manuscript.**
> We have updated the "Limitations and Future Directions" section to transparently discuss this aspect. We acknowledge that while our data is expertly curated, it may not perfectly capture the long-tail distribution of real user requests. We position this work as a foundational step that paves the way for future research involving naturalistic data collection (e.g., user logs).
>
> [1] Yizhong Wang, Yeganeh Kordi, Swaroop Mishra, Alisa Liu, Noah A. Smith, Daniel Khashabi, and Hannaneh Hajishirzi. Self-instruct: Aligning language models with self-generated instructions, 2023.
>
> [2] Wei Liu, Yancheng He, Hui Huang, Chengwei Hu, Jiaheng Liu, Shilong Li, Wenbo Su, and Bo Zheng. Air: Complex instruction generation via automatic iterative refinement, 2025.

---

> ### Author Response · Authors · 2025-11-20
> **Rebuttal by Authors**
>
> ``Question 2(Q2)``:
> > Are there plans to support multi-turn instruction-following, where users refine their requests iteratively?
>
> We appreciate the reviewer raising this critical direction for future research.
>
> Multi-turn interaction, where a user iteratively refines their request, represents a more complex and realistic use case for instruction-following models. While our current single-turn benchmark, IF-VidCap, is designed to systematically evaluate a wide range of atomic constraints, developing a new benchmark for multi-turn video instruction following is indeed an exciting and logical extension. We believe this constitutes a substantial research effort in its own right and plan to systematically and thoroughly investigate benchmarking for multi-turn instruction following in our future work.

---

> ### Author Response · Authors · 2025-11-24
> **Update: Experimental Results on Long Videos (1-10m)**
>
> Thank you very much for your patience. To specifically address your concerns regarding long-video evaluation, we conducted supplementary experiments and verifications targeting this temporal range during the rebuttal phase.
>
> To validate the effectiveness of our evaluation framework in long-video scenarios, we constructed **600 test samples** covering durations of **1 to 10 minutes**, based on 150 video sources.
>
> Unlike the previous short-video section, to align with practical application scenarios for long videos, we incorporated instruction requirements specific to long contexts into the controlled description tasks, such as **Cross-scene description**, **Focused-scene description**, and **Script-structure description**. It is worth noting that these new instructions are constructed entirely based on our original 27 atomic constraint system without adding new atomic types, ensuring consistency in evaluation standards.
>
> We tested the most representative closed-source and open-source models (covering different parameter scales), and the results are as follows:
>
> **Table 1: Experimental Results on Long-Video Dataset (1min-10min)**
> | Models | Params | Overall ISR | Overall CSR | Rule-based ISR | Rule-based CSR | Open-ended ISR | Open-ended CSR |
> | :--- | :---: | :---: | :---: | :---: | :---: | :---: | :---: |
> | *Closed-Source Models* | | | | | | | |
> | **Gemini-2.5-Pro** | 🔒 | 25.45 (↓2.38) | 71.20 (↓3.33) | 71.50 | 84.10 | 29.80 | 52.50 |
> | **Gemini-2.5-Flash** | 🔒 | 21.80 (↓3.7) | 68.10 (↓4.53) | 65.90 | 81.50 | 25.50 | 48.20 |
> | **GPT-4o** | 🔒 | 14.50 (↓8.4) | 59.30 (↓11.44)| 67.78 | 82.40 | 20.10 | 40.80 |
> | *Open-Source Models* | | | | | | | |
> | **InternVL-3.5** 💡 | 38B | 15.42 / 12.20 | 66.10 / 59.50 | 55.20 / 48.50 | 74.10 / 70.20 | 17.23 / 12.36 | 38.40 / 30.10 |
> | **Qwen2.5-VL-Instruct**| 72B | 12.80 (↓4.7) | 58.50 (↓8.78) | 53.40 | 73.10 | 14.90 | 36.20 |
> | **Qwen2.5-VL-Instruct**| 32B | 9.75 (↓5.41) | 53.10 (↓10.94)| 47.10 | 68.50 | 11.20 | 32.50 |
> | **InternVL-3.5** 💡 | 8B | 8.95 / 3.80 | 54.20 / 42.10 | 42.60 / 28.50 | 65.80 / 53.40 | 10.15 / 3.60 | 30.60 / 20.10 |
> | **Qwen2.5-VL-Instruct**| 7B | 4.80 (↓6.12) | 46.50 (↓11.62)| 32.50 | 58.20 | 6.50 | 23.80 |
>
> As shown in Table 1, we have several key findings from the long-video experimental results:
>
> 1. **Significantly increased difficulty in instruction following under long temporal contexts:**
> Almost all models showed varying degrees of decline (↓) in Instruction Satisfaction Rate (ISR) compared to short-video tasks. Notably, GPT-4o (ISR 14.50) dropped by 8.4 points. This indicates that when processing visual contexts spanning up to 10 minutes, models face a difficult trade-off between "understanding the macro-narrative" and "adhering to fine-grained constraints," often failing to attend to both.
> 2. **Reasoning (Thinking) demonstrates advantages surpassing parameter size in long videos:**
> This is the most surprising finding. **InternVL-3.5-38B (Thinking)**, with thinking mode enabled, achieved an ISR of 15.42, not only outperforming the larger Qwen2.5-VL-72B (12.80) but also surpassing the closed-source model GPT-4o (14.50).
> This strongly proves that in complex structured description tasks for long videos (such as script generation), introducing the Chain-of-Thought (Thinking) mechanism is more effective than simply increasing model parameters or expanding the context window. The thinking process helps the model organize fragmented information extracted from the long video before generating the final response, thereby better satisfying the constraints.
> 3. **Small models face the challenge of "long-context forgetting," but Thinking mode offers a buffer:**
> For 7B/8B level models, processing 10-minute videos is extremely challenging. The ISR of Qwen2.5-VL-7B and InternVL-8B (Non-thinking) dropped to single digits (4.80 and 3.80), indicating severe instruction following breakdown. However, after enabling the Thinking mode, InternVL-8B's ISR improved to 8.95, nearly reaching the level of Qwen-32B. This further verifies that reasoning computation can partially compensate for the lack of parameter scale.
> 4. Robustness of the Gemini architecture:
> The Gemini-2.5 series showed the least performance degradation (Pro version ISR dropped by only 2.38). This is likely due to its architectural advantage of natively supporting ultra-long contexts, allowing it to maintain more stable instruction-following capabilities across durations from 1 to 10 minutes.
>
> In summary, these supplementary experiments not only positively validate the applicability of our evaluation metrics in long-video scenarios but also reveal important directions for optimizing long-video understanding models—namely, the combination of "long-context architecture" and "reasoning capabilities." These findings strongly support our conclusions regarding the fine-grained capabilities of the models.

---

> ### Comment · Reviewer_z26H · 2025-11-27
>
> Thank you for your substantial responses. I maintain my original rating.

---

> > ### Author Response · Authors · 2025-11-27
> >
> > Dear Reviewer z26H,
> >
> > Thank you again for your time and for acknowledging our response as substantial.
> >
> > To ensure we have fully met your expectations and to contribute the most reliable benchmark to the community, **we would like to briefly summarize how we have addressed your specific concerns**:
> >
> > 1. **Long-Video Gap**: We successfully extended the benchmark to the 1-10 minute range (600 new samples), revealing that "Thinking/Reasoning" models can outperform larger models in long-context tasks.
> > 2. **Fluency & Quality**: We demonstrated via basic experiments that dedicated fluency evaluation is unnecessary, aligning with industry consensus on the saturation of current LLM capabilities. This validates our benchmark's design choice to focus strictly on verifiable instruction adherence rather than generic text quality.
> > 3. **Data Scale**: We clarified that our original scale (1.4K) already exceeds comparable text benchmarks (e.g., IFEval), and the addition of the long-video subset (600 new samples) has further expanded this scale advantage.
> > 4. **Training Data Distribution**: We clarified that our "caption-to-instruction" method aligns with established paradigms like Self-Instruct, suitable for emerging domains. Crucially, we detailed our human supervision process, where the final curated data achieved high scores in "Naturalness" (4.63/5.0) and "Fidelity" (4.76/5.0) via iterative refinement, bridging the gap with real user queries.
> >
> > **Given that we have addressed all the weaknesses and questions raised in your initial review (W1-W3&Q1-Q2), could you kindly let us know if you still have any specific concerns?**
> >
> > We are fully prepared to answer any further questions immediately to clear up any hesitations. Our goal is to ensure IF-VidCap serves as a robust foundation for future research.
> >
> > Best regards,
> > The Authors

---

### Author Response · Authors · 2025-11-29
**General Response to All Reviewers**

We thank all reviewers for their time and effort in reviewing our manuscript, "IF-VidCap: Can Video Caption Models Follow Instructions?". We are grateful for the insightful and constructive comments. We believe that by addressing these issues, the quality and clarity of IF-VidCap have been significantly improved. The major updates and responses are summarized as follows:

1.**Clarification and Extension of Video Duration:** We have clarified the rationale for targeting the 2-60s video duration range (avg. 20.5s). Addressing reviewers' concerns, we have expanded our benchmark to include long-video testing and reported the corresponding results (See Reviewer z26H W1&Q1, mY9K W1, h647 W3).

2.**Dataset Quality, Diversity, and Distribution:** We have included a discussion on the correlation and similarity between the training and test sets. We further clarified the diversity and quality control of our "caption-to-instruction" training set construction pipeline, demonstrating its alignment with real-world user instruction distributions (See Reviewer z26H W3, mY9K Q4, sjFE Q2).

3.**Scope of the Benchmark:** We have clarified that our benchmark primarily focuses on explicitly verifiable instruction following, rather than abstract, open-ended instructions. We also discussed the challenges and significance of extending evaluation to open-ended domains (See Reviewer z26H W2, mY9K Q7).

4.**Additional Implementation Details and Analysis**: We have provided more details and answers to specific questions raised in the initial review, including consistency control in the annotation process, error type analysis, and detailed interpretations of figures and charts (See Reviewer mY9K Q2&Q3, h647 W1&W2, sjFE W4).

5.**Human Performance and Expanded Baselines:** To better contextualize the difficulty of our benchmark, we have added a human-level performance baseline and evaluated closer-sourced models. These results have been updated in the revised manuscript (See Reviewer sjFE W1&W2&Q1).

6.**Reliability of Auto-Evaluation (LLM-as-a-judge):** Addressing concerns about the consistency between LLM judges and human evaluators, we conducted further alignment tests on CSR/ISR metrics. The results demonstrate that advanced evaluation models achieve high agreement with human evaluators, validating the reliability of our metrics (See Reviewer mY9K Q6).

7.**Openness and Reproducibility:** We emphasize that both the benchmark dataset and the evaluation toolkit code have been released to the community to ensure reproducibility and facilitate future research (See Reviewer mY9K Q1, h647 W4).

8.**Comparison with Traditional Metrics:** We have added an analysis of standard unconstrained captioning metrics (BLEU, CIDEr, CLIPScore) to provide a comprehensive comparison with our task-specific metrics (See Reviewer mY9K Q5).

9.**Generalization and Overfitting Analysis:** We evaluated our fine-tuned model on external video captioning benchmarks (VidCapBench and Dream-1K). Detailed comparisons with baseline models demonstrate that our instruction-tuning improves descriptiveness and comprehensiveness without overfitting (See Reviewer sjFE Q3).

---

### Meta-Review · Area_Chair_kvco · 2026-01-06

**Summary:**

The reviewers consistently acknowledge the strengthes and contributions of this work. During the rebuttal,  the authors addressed several core concerns by extending the benchmark to long videos (1–10 minutes), adding a human performance baseline, expanding baseline coverage, and providing extensive analyses on constraint types, data quality, and evaluation reliability. With these substantial improvements, the major concerns raised in the initial reviews have been effectively resolved. I therefore recommend acceptance of this submission.

**Reviewer Concerns:**

During the rebuttal, the authors have addressed several core concerns by extending the benchmark to long videos (1–10 minutes), adding a human performance baseline, expanding baseline coverage, and providing extensive analyses on constraint types, data quality, and evaluation reliability. The authors also released datasets, prompts, and evaluation code to ensure reproducibility.

**Reviewer Scores:**

The initial reviewer scores are 4/4/8/8. By carefully reviewing the comments and the authors’ responses, I believe the reviewers would be inclined to maintain their original ratings or potentially increase their scores since their major concerns have been solved.

---

### Decision · Program_Chairs · 2026-01-26

Accept (Poster)